# Radon and Its Short-Lived Products in Indoor Air: Present Status and Perspectives

**Janja Vaupotič**

Department of Environmental Sciences, Jožef Stefan Institute, Jamova cesta 39, SI-1000 Ljubljana, Slovenia; janja.vaupotic@ijs.si

**Abstract:** Initially, basic equations are given to express the activity concentrations and concentrations of potential $\alpha$-energies of radon ($^{222}$Rn) and thoron ($^{220}$Tn) and their short-lived products in indoor air. The appearance of short-lived products as a radioactive aerosol is shown, and the fraction of the unattached products is particularly exposed, a key datum in radon dosimetry. This fundamental part is followed by giving the sources of radon and thoron indoors, and thus, their products, and displaying the dependence of their levels on the ground characteristics, building material and practice, and living–working habits of residents. Substantial hourly, daily, and seasonal changes in their activity concentrations are reviewed, as influenced by meteorological parameters (air temperature, pressure, humidity, and wind speed) and human activity (either by ventilation, air conditioning and air filtration, or by generating aerosol particles). The role of the aerosol particle concentration and their size distribution in the dynamics of radon products in indoor air has been elucidated, focusing on the fraction of unattached products. Intensifying combined monitoring of radon short-lived products and background aerosol would improve radon dosimetry approaches in field and laboratory experiments. A profound knowledge of the influence of meteorological parameters and human activities on the dynamics of the behaviour of radon and thoron accompanied by their products in the air is a prerequisite to managing sustainable indoor air quality and human health.

**Keywords:** radon; thoron; short-lived products; aerosol; relationship; indoor

## 1. Introduction

In the three primordial radioactive chains, three radon isotopes are created by $\alpha$-transformation of radium [1]: $^{222}$Rn (half-life, $t_{1/2}$ = 3.82 days) in the uranium chain starting with $^{238}$U, $^{220}$Rn ($t_{1/2}$ = 55.6 s) in the thorium chain starting with $^{232}$Th, and $^{219}$Rn ($t_{1/2}$ = 3.9 s) in the actinium chain starting with $^{235}$U (Figure 1). Only a fraction of radon atoms (emanation fraction or emanation power) succeed in leaving the mineral grain due to their recoil energy and thus enter the void space, from where they migrate through the medium either by diffusion or, more effectively and to longer distances, by being carried by gas or water [2]. On its way, radon accumulates in underground rooms (e.g., fissures, karst caves, mines, basements) and, eventually, exhales in the atmosphere and appears in the air of living and working environments. Generally, the predominant $^{222}$Rn source for indoor activity concentration is its level in the ground on which a building stands [3–5]. Only seldom water or natural gas used in a household could be the primary source [6,7]. On the other hand, because of its short half-life, $^{220}$Rn cannot travel far from its source, and, therefore, it appears indoors at radiologically relevant levels mostly only when the floor, walls, or other indoor elements are made of material of elevated thorium content [8,9]. Only exceptionally high thoron levels in soil [10] or in outdoor air [11] can result in high levels in indoor air. $^{219}$Rn, with its very short half-life, is never met at relevant levels in ambient air. Hereafter, $^{222}$Rn is also referred to as radon and $^{220}$Rn is referred to as thoron.

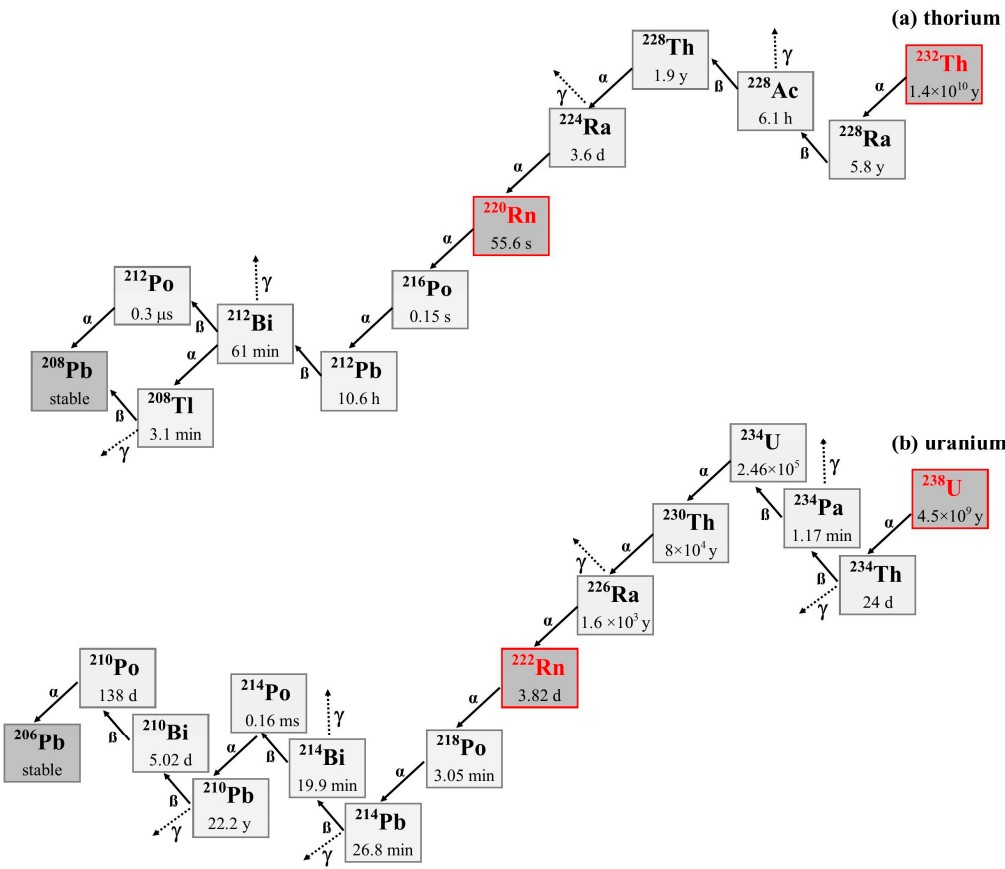

**Figure 1.** Radioactive chains of (**a**) thorium ($^{232}$Th) and (**b**) uranium ($^{238}$U); adapted by the World Nuclear Association, http://www.world-nuclear.org/information-library/safety-and-security/radiation-and-health/naturally-occurring-radioactive-materials-norm.aspx (accessed on 21 March 2023).

Radon and thoron $\alpha$-transformations are followed by radioactive chains of their short-lived metallic products (Figure 1). Radon (RnP) and thoron (TnP) products are initially mostly positive ions [12–14], which soon react with molecules of trace gases and vapours (mostly water) in air and are partly oxidised and form small charged and neutral clusters [13–15]. These are considered unattached RnP and TnP. Product ions and clusters also attach in part to the background aerosol particles. The extent of attachment depends on environmental conditions [13,16–18]. These products are denoted as attached RnP and TnP. Thus, RnP and TnP appear as radioactive aerosols. When breathing, aerosol particulates are partly deposited on the walls of the respiratory tract. It has been recognised that on average, globally, inhaled RnP and TnP contribute about half of the effective dose (the contribution of radon and thoron gas being minor) a member of the general public receives from all-natural radioactivity [19], and they are a major cause of lung cancer, second only to cigarette smoking [20].

Systematic and extensive radon measurements were first introduced and conducted in the uranium mines and mills in the uranium era, soon after the Second World War, as a means to maintain ventilation at workplaces sufficiently effective to ensure exposure to radon under acceptably low levels, which have been steadily lowering, under increased radiation protection concern for workers. Towards the end of the 20th century, it became clear that elevated radon levels could also appear in the indoor air of dwellings. As a result, international institutions responsible for health care and radiation protection of the general population started and have continued to publish recommendations that limit radon levels in the living and working environment, thus stimulating national governments to issue their related legislation. As a result, systematic and often extensive radon surveys in dwellings

have been initiated and conducted in a large number of countries, mainly in Europe, the United States, and India [19]. In some countries, following the recommendations of the US Environmental Protection Agency [21,22], the national radon survey was first carried out in kindergartens and schools, as reviewed by Vaupotič [23], with a young population more vulnerable to radiation [24–27], and then continued in residential buildings. In his comprehensive review, Porstendörfer [11] reported and commented on the results of 149 works, only 27 of which were devoted to thoron. In another paper [28], thoron results were reviewed in 15 studies conducted in 10 countries.

In the beginning, priority was given to radon because of a general opinion that thoron, due to its short half-life, was less likely to accumulate at high levels indoors. This opinion was slowly changed as more papers appeared reporting thoron levels comparable to or even higher than radon levels [29], such as in traditional wooden Japanese houses [30,31], in Italian buildings made of volcanic material [8], or in cave dwellings in China [32]. Nonetheless, extensive monitoring of thoron and its products was not possible until recently, when the appropriate solid-state nuclear track detectors became commercially available. In many countries that had already finished their national radon programmes, thoron was measured recently, at least to a limited extent, to estimate its indoor levels and complement their previous radon database, e.g., Canada [33], Ireland [34], Germany [35], Poland [36], Hungary [37], Slovenia [38], Serbia and Kosovo [39,40], and North Macedonia [41]. On the other hand, extensive studies of radon, thoron, and their products have recently been conducted in various countries, particularly in several areas in India. Some solely discuss radon and thoron in indoor air, soil, and water [42–44], while others discuss radon, thoron, and their progeny in dwellings in different parts of the country [45,46], including thoron dose estimates [47–50], or are focused on measurement technique [51,52]. Nonetheless, the thoron database remains modest in comparison to that of radon.

The local and national radon surveys have generally been accompanied by radiation dose estimates and followed by radon mitigation in buildings with radon levels above a certain limit. In order to harmonise the results obtained in field measurements by different groups, several international intercomparison experiments have been organised. A significant aim has been achieved by the European Radon Atlas, to which a majority of the European countries have contributed with their indoor air and soil gas radon databases.

## 2. Fundamentals of Radon and Thoron Products

Figure 2 presents radon and thoron sub-chains. $^{218}$Po and $^{216}$Po are formed as positively charged and become neutralised in one of the following processes [13–15]: (i) recombination with ions produced by $\alpha$, $\beta$, and $\gamma$ emissions and recoil atoms during radioactive transformations of airborne radionuclides, as well as by background $\gamma$ and cosmic rays, (ii) electron scavenging by OH radicals formed by radiolysis of water molecules, and (iii) charge transfer from molecules of lower ionisation potential. The states of $^{218}$Po, $^{216}$Po, and their oxides at the moment of their $\alpha$-transformation into Pb are decisive because they determine the initial characteristics and behaviour of the subsequent members in the chains. For $^{218}$Po in 50% humid air at an ionisation rate of 3.2 pC kg$^{-1}$ s$^{-1}$ (45 μR h$^{-1}$), the rate constants of the above processes of neutralisation are $0.07 \times 10^{-2}$ s$^{-1}$, $1.07 \times 10^{-2}$ s$^{-1}$, and $0.4 \times 10^{-2}$ s$^{-1}$, respectively, and, eventually, more than half of the species (molecular clusters) are neutral [16]. The clusters of RnP and TnP species are called unattached RnP and TnP. The above processes are accompanied and followed by the attachment of clusters [13,16–18,53], both charged and already neutralised, to the background aerosol particles, thus forming attached RnP and TnP. The attachment rate constant $\lambda_a$ (s$^{-1}$) is proportional to the number concentration of aerosol particles (number of particles in the volume unit, $N$/cm$^{-3}$) and the size-dependent [18] attachment probability (coefficient $\beta$/cm$^3$ s$^{-1}$), expressed as [54]:

$$\lambda_a = \beta N. \tag{1}$$

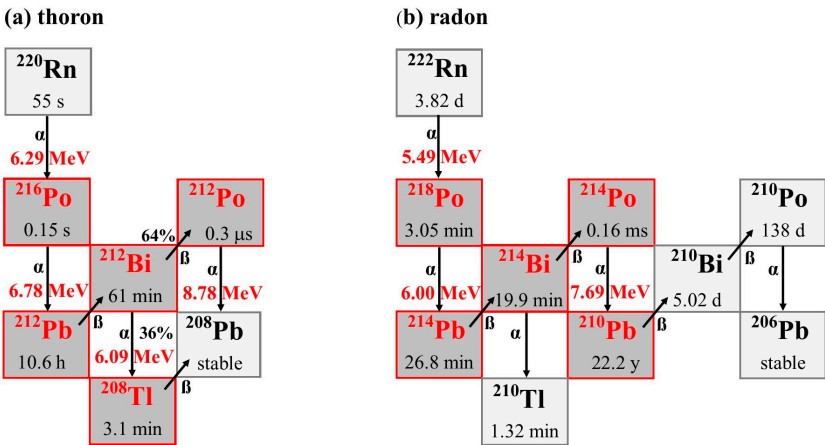

**Figure 2.** Radioactive sub-chains of (**a**) thoron ($^{220}$Rn) and (**b**) radon ($^{222}$Rn).

Because of the recoil energy (order of magnitude 100 eV) gained during $\alpha$-transformation of the attached $^{218}$Po and $^{216}$Po, a considerable recoil fraction ($r$) of $^{214}$Pb and $^{212}$Pb atoms is ejected from the aerosol particles [55,56]. Attached RnP and TnP species behave as other radioactive and non-radioactive aerosol particles. As such, they are also subject to steady particle deposition. The deposition rate constant $\lambda_d$ (s$^{-1}$) depends on the particle deposition velocity $v_d$ (m s$^{-1}$), corrected for friction velocity [57,58], and is proportional to the ratio of the available surface $S$ (m$^2$) in a room of volume $V$ (m$^3$) [56]:

$$\lambda_d = v_d \frac{S}{V}. \tag{2}$$

The deposition rate constant $\lambda_d$ is distinguished for the unattached ($\lambda_d^u$) and attached ($\lambda_d^a$) species. Instead of deposition velocity $v_d$ (Equation (2)), some authors use the so-called average or effective deposition velocity ($v_d^{eff}$), expressed by the deposition velocity of the unattached ($v_d^u$) and attached ($v_d^a$) species as [57,59]:

$$v_d^{eff} = f^u v_d^u + (1 - f^u) v_d^a, \tag{3}$$

with $f^u$—a fraction of unattached RnP or TnP.

Potential $\alpha$-energies associated with RnP and TnP transformations (Figure 2) are summarised in Table 1 [56,60–63]. To each isotope, all energies are assigned and emitted in its transformations along the chain to the stable $^{206}$Pb for RnP and stable $^{208}$Pb for TnP. Thus, the $^{218}$Po atom has (6.00 + 7.69) MeV, and $^{214}$Pb and $^{214}$Bi atoms (though only $\beta$ emitters) have 7.69 MeV. Because the TnP chain is branched after $^{212}$Bi, the average value of 7.81 MeV (=0.36 × 6.09 + 0.64 × 8.78) is assigned to $^{212}$Pb and $^{212}$Bi, and (6.78 + 7.81) MeV is assigned to $^{216}$Po. Energies are given per one atom and per 1 Bq (i.e., $E_\alpha$ and $E_\alpha/\lambda$, $\lambda$—rate constant of radioactive transformation). For RnP, the summation of $(E_\alpha/\lambda)_j$, with j = 1 to 4, gives 34,520 MeV. To this sum, a fraction of $k_j = (E_\alpha/\lambda)_j/34,520$ (Table 1) is contributed by each of RnP species, with a fraction of $^{218}$Po being negligible. Thus, the above situation represented as an imaginary secular equilibrium condition in which activities of all of the products are equal to the radon activity, i.e., 1 Bq, can be written as

$$A_{RnP} = 0.106 A_{218Po} + 0.515 A_{214Pb} + 0.379 A_{214Bi} + 6 \times 10^{-8} A_{214Po}, \tag{4}$$

with $A_{RnP}$ (Bq m$^{-3}$), thus expressing the equilibrium-equivalent activity concentration of RnP (also denoted by EERC, EECRn, or, rarely, $C_{eq,Rn}$).

**Table 1.** Energetics of radon ($^{222}$Rn) and thoron ($^{220}$Rn) and their short-lived products: $\lambda$—rate constant of radioactive transformation ('decay constant'), $t_{1/2}$—half-life (ln 2/$\lambda$), $E_\alpha$—potential $\alpha$-energy of radioactive transformation.

| Radionuclide | j | $t_{1/2}$ | $E_\alpha$ per 1 Atom | | $\frac{E_\alpha}{\lambda}$ per 1 Bq | | |
| | | | MeV | pJ | MeV | nJ | $k_j$ |
|---|---|---|---|---|---|---|---|
| $^{222}$Rn | 0 | 3.82 d | 19.18 | 3.07 | $9.2 \times 10^6$ | 147 | / |
| $^{218}$Po | 1 | 3.05 min | 13.69 | 2.19 | 3620 | 0.579 | 0.106 |
| $^{214}$Pb | 2 | 26.8 min | 7.69 | 1.23 | 17,800 | 2.86 | 0.515 |
| $^{214}$Bi | 3 | 19.9 min | 7.69 | 1.23 | 13,100 | 2.1 | 0.379 |
| $^{214}$Po | 4 | 164 µs | 7.69 | 1.23 | $2 \times 10^{-3}$ | $2.9 \times 10^{-5}$ | $6 \times 10^{-8}$ |
| $\sum\limits_{j=1}^{4} (E_\alpha/\lambda)_j$ | | | | | $34.52 \times 10^3$ | | |
| $^{220}$Rn | 0 | 55.6 s | 20.88 | 3.34 | 1660 | 0.265 | / |
| $^{216}$Po | 1 | 0.15 s | 14.59 | 2.33 | 3.32 | $5.3 \times 10^{-4}$ | $7 \times 10^{-6}$ |
| $^{212}$Pb | 2 | 10.6 h | 7.81 | 1.25 | $4.3 \times 10^5$ | 6.91 | 0.913 |
| $^{212}$Bi | 3 | 60.5 min | 7.81 | 1.25 | $4.1 \times 10^4$ | 6.56 | 0.087 |
| $^{212}$Po | 4 | 299 ns | 8.78 | 1.41 | $3.9 \times 10^{-6}$ | $6.9 \times 10^{-10}$ | $8 \times 10^{-12}$ |
| $\sum\limits_{j=1}^{4} (E_\alpha/\lambda)_j$ | | | | | $47.1 \times 10^4$ | | |

$k_j$ values for RnP and TnP sub-chains are calculated by: $k_j = \left(\frac{E_\alpha}{\lambda}\right)_j / \sum\limits_{j=1}^{4} \left(\frac{E_\alpha}{\lambda}\right)_j$.

If 3700 Bq m$^{-3}$ (100 pCi m$^{-3}$) is taken instead of 1 Bq m$^{-3}$, the concentration of potential $\alpha$-energy is $1.29 \times 10^8$ MeV m$^{-3}$ (20.4 µJ m$^{-3}$), which is equivalent to the RnP concentration of 1 WL ('working-level'—an early radon limit for uranium miners), regardless of the degree of secular equilibrium between Rn and RnP.

By multiplying $E_\alpha/\lambda$ values from Table 1 with the activity concentrations of the related radionuclides, the concentration of potential $\alpha$-energy of RnP is obtained ($E_{\alpha RnP}$/MeV m$^{-3}$, also denoted as PAEC or PAEC$_{Rn}$) as follows:

$$E_{\alpha RnP} = 3620 A_{218Po} + 17800 A_{214Pb} + 13100 A_{214Bi} + 2 \times 10^{-3} A_{214Po}. \tag{5}$$

Referring to the energetics of TnP in Table 1 and applying the same reasoning as above for RnP, the equilibrium-equivalent activity concentration of TnP ($A_{TnP}$) (also denoted by EETC, EECTn, or, rarely, $C_{eq,Tn}$) is expressed as

$$A_{TnP} = 7 \times 10^{-6} A_{216Po} + 0.913 A_{212Pb} + 0.087 A_{212Bi} + 8 \times 10^{-12} A_{212Po}, \tag{6}$$

and the concentration of potential $\alpha$-energy of TnP ($E_{\alpha TnP}$/MeV m$^{-3}$, also denoted as PAEC or PAEC$_{Tn}$) is expressed as:

$$E_{\alpha TnP} = 3.32 A_{216Po} + 4.3 \times 10^5 A_{212Pb} + 4.1 \times 10^4 A_{212Bi} + 3.9 \times 10^{-6} A_{212Po}. \tag{7}$$

As 1 Bq of TnP contains $47.1 \times 10^4$ MeV (Table 1) and 1 WL is defined as $1.29 \times 10^8$ MeV m$^{-3}$, 1 WL of TnP corresponds to $A_{Tn} = 275$ Bq m$^{-3}$ (=$1.29 \times 10^8$ MeV m$^{-3}$/$47.1 \times 10^4$ MeV Bq$^{-1}$).

Exposure of 1 WLM ('working-level-month') is gained by breathing air of radon or thoron activity concentration of 1 WL for 170 h.

Due to all of the sinks of radon and thoron products (e.g., deposition, filtration), secular equilibria between radon and its products and thoron and its products are never reached in indoor and outdoor air, and the actual degree of equilibrium is expressed by the so-called equilibrium factor $F$, defined as:

$$F_{Rn} = A_{RnP}/A_{Rn} \text{ for radon} \tag{8}$$

and

$$F_{\text{Tn}} = A_{\text{TnP}}/A_{\text{Tn}} \text{ for thoron.} \tag{9}$$

Sometimes, when $F_{\text{Rn}}$ is measured with the solid-state nuclear track detectors, the so-called reduced equilibrium factor, defined as [64,65]

$$F_{\text{Rn}}^{\text{red}} = (0.106 A_{218\text{Po}} + 0.379 A_{214\text{Bi}})/A_{\text{Rn}}, \tag{10}$$

or the proxy equilibrium factor, defined as [65]

$$F_{\text{Rn}}^{\text{p}} = (A_{218\text{Po}} + A_{214\text{Bi}})/A_{\text{Rn}}, \tag{11}$$

are reported and may, under certain conditions, approximate the total equilibrium factor $F_{\text{Rn}}$ well.

Dependence of particle deposition on their size [66,67] plays a crucial role in modelling the adsorption of aerosol particles (and thus RnP and TnP) to the walls of the respiratory tract along the lung generations during breathing. This is a prerequisite for radon dosimetry because it enables us to calculate dose conversion factors $f_{\text{DC}}$ (also denoted as DCF), expressing radiation dose (Gy or Sv) per unit exposure (Bq m$^{-3}$ h or WLM) [68–73]. Birchall and James [72] and Marsh et al. [73] have shown that the parameter mainly affecting $f_{\text{DC}}$ is the fraction of unattached radon and thoron products ($f^{\text{u}}$), defined as [1]

$$f_{\text{RnP}}^{\text{u}} = \frac{A_{\text{RnP}}^{\text{u}}}{A_{\text{RnP}}} \text{ or } f_{\text{RnP}}^{\text{u}} = \frac{E_{\alpha\text{RnP}}^{\text{u}}}{E_{\alpha\text{RnP}}} \text{ for RnP} \tag{12}$$

and

$$f_{\text{TnP}}^{\text{u}} = \frac{A_{\text{TnP}}^{\text{u}}}{A_{\text{TnP}}} \text{ or } f_{\text{TnP}}^{\text{u}} = \frac{E_{\alpha\text{TnP}}^{\text{u}}}{E_{\alpha\text{TnP}}} \text{ for TnP,} \tag{13}$$

where $A_{\text{RnP}}^{\text{u}}$, $A_{\text{TnP}}^{\text{u}}$, $E_{\alpha\text{RnP}}^{\text{u}}$, and $E_{\alpha\text{TnP}}^{\text{u}}$ are obtained if in Equations (4)−(7) the activity concentrations of only the unattached species of the individual products are included, and not their total concentrations (e.g., $A_{218\text{Po}}^{\text{u}}$ instead of $A_{218\text{Po}}$). Empirical formulae have been proposed to use $f_{\text{RnP}}^{\text{u}}$ to calculate $f_{\text{DC}}$ (in mSv WLM$^{-1}$) for RnP [72,73]

$$f_{\text{DC}} = 11.35 + 43 f_{\text{RnP}}^{\text{u}}, \tag{14}$$

and separately for

$$\text{nasal breathing}: \ f_{\text{DC}} = 101 f_{\text{RnP}}^{\text{u}} + 6.7(1 - f_{\text{RnP}}^{\text{u}}) \tag{15}$$

and [74]

$$\text{mouth breathing}: \ f_{\text{DC}} = 23 f_{\text{RnP}}^{\text{u}} + 6.2(1 - f_{\text{RnP}}^{\text{u}}). \tag{16}$$

As reviewed by Porstendörfer and Reineking [14], $f_{\text{RnP}}^{\text{u}}$ differs substantially from place to place, and depending on the environmental conditions, its value ranges from 0.006 to 0.83. Generally, it is inversely proportional to the number concentration of aerosol particles [75–78], the relationship being approximated by Papastefanou [18], Porstendörfer [74], and Huet et al. [79]:

$$f_{\text{RnP}}^{\text{u}} = \frac{400}{N/\text{cm}^{-3}}. \tag{17}$$

Thus, $f_{\text{Rn}}^{\text{u}}$ is very low in mines with high aerosol concentration [75] and high in karst caves with very clean air [75,76,80–83].

The related empirical relation for thoron products is [18]:

$$f_{\text{TnP}}^{\text{u}} = \frac{150}{N/\text{cm}^{-3}}. \tag{18}$$

## 3. Radon Short-Lived Products as Radioactive Aerosols

Ambient air is a suspension of fine solid particles or liquid droplets. Particles are of different sizes, shapes, and physical–chemical properties. A special class is radioactive aerosols made up of radioactive particles, both as molecular clusters of radionuclides and radionuclides attached to particulate matter [18]. Radon and thoron products belong to this class. The size of aerosol particles ranges from less than several nm for molecular clusters to about 100 μm for fog droplets and dust particles. Particles larger than 100 μm cannot remain suspended in air and may not, therefore, be considered as aerosol [84]. Classification of aerosols with respect to their particle size differs depending on the purpose of use, as well as the author [84–88]. Terms of ultrafine (<100 nm), fine (<1000 nm), and coarse particles (>1000 nm) are preferably used by toxicologists and regulatory bodies. On the other hand, aerosol scientists refer mainly to different modes [89]: nucleation (1–30 nm), Aitken (20–100 nm), and accumulation mode (90–1000 nm). Nevertheless, the borders are not strictly fixed and may differ from author to author. Neither is the term nano applied univocally. Although it may refer to any particle of <1 μm size, it is used for particles of <300 nm [86], <100 nm [85,90], <50 nm [91], or even smaller [92,93].

Monodisperse aerosols are very rare. The size distribution of aerosol particles is mathematically described either by differential or integral distribution function, usually in logarithmic form. Number size distribution $P_{\mathrm{L}}^{N}(d)$, showing the number concentration of particles (in $\mathrm{m}^{-3}$ or, often, more conveniently, in $\mathrm{cm}^{-3}$ or even $\mathrm{mm}^{-3}$) within the size windows of particle diameters $d$ over the entire size range is expressed in the logarithmic form as [88]:

$$P_{\mathrm{L}}^{N}(d) = -\frac{\mathrm{d}N}{\mathrm{d}(\ln d)}. \tag{19}$$

For some purposes, the distribution of the concentration of mass, surface, or volume of particles (instead of the number) with respect to the particle size can be useful [18]. For radon and thoron products, as for other radioactive aerosol particles [18,53], the activity size distribution $P_{\mathrm{L}}^{A}(d)$ is commonly used:

$$P_{\mathrm{L}}^{A}(d) = -\frac{\mathrm{d}A}{\mathrm{d}(\ln d)}. \tag{20}$$

The two distributions are related as [54,74,94]:

$$P_{\mathrm{L}}^{A}(d) = \frac{A_{\mathrm{r}}}{\lambda_{\mathrm{a}}}\beta(d)P_{\mathrm{L}}^{N}(d), \tag{21}$$

with r—radionuclide in question.

For the activity size distribution, the activity median diameter (AMD) or geometric mean diameter (GMD) is reported. It is named AMAD (activity median aerodynamic diameter) when the impactor is used for measurement or AMTD (activity median thermodynamic diameter) when the diffusion battery is used [95].

Measurements in 31 occupied houses in New Jersey, USA showed a bimodal activity size distribution with GMD ranging from $1\times/:1.38$ nm to $9\times/:2.61$ nm for unattached and from $40\times/:6.30$ nm to $258\times/:1.67$ nm for attached RnP [96]. In another study, the same authors [95] showed that bimodal size distributions of RnP and TnP particles are similar and that they do not differ significantly even in a range of the aerosol particle number concentration from 2.3 to 180 $\mathrm{mm}^{-3}$ (Table 2). On the other hand, depending on the aerosol source and its concentration, uni-, bi-, and three-modal activity size distribution of RnP was observed [97].

Hopke et al. [98] considered unattached RnP as particles in the size range of 0.5–1.5 nm. According to a review by Porstendörfer and Reineking [14], AMD of the RnP clusters falls into the range of 0.9 nm to 30 nm, while AMAD of the aerosol particles that carry RnP attached are in the range of 50 nm to 500 nm. Measurements in indoor air also showed that within the unattached region of <10 nm, two (with AMD of 0.8 and 4.2 nm) or even

three activity distribution peaks (0.60, 0.85, and 1.25 nm) may appear [74,99]. In addition, attached RnP appeared in the nucleation (attached to particles of 14–40 nm), accumulation (210–310 nm), and coarse mode (3000–5000 nm) [99]. In an intercomparison experiment carried out in a test chamber, the following average AMD values and regions (nm) were found for the unattached RnP [100]: 0.66 (0.53–0.86) for $^{218}$Po, 0.80 (0.53–1.76) for $^{214}$Pb, and 0.91 (0.53–2.20) for $^{214}$Bi. The following were found for TnP: 0.73 (0.53–0.97) for $^{212}$Pb and 0.85 (0.53–2.31) for $^{212}$Bi. According to Huet et al. [101], diameters (nm) of the unattached $^{218}$Po, $^{214}$Pb, and $^{214}$Bi in an aged aerosol were similar and close to $0.85\times/{:}1.25$, and of the attached RnP, to $190\times/{:}1.64$. In a radon chamber containing carrier aerosol, the AMD values of 0.82, 0.79, 1.70, and 0.82 nm were obtained for the unattached $^{218}$Po, $^{214}$Pb, $^{214}$Bi, and $^{214}$Po, respectively [102]. For unattached $^{212}$Pb, the diameter size range of 0.8–2.0 nm and AMTD of 1.32 nm have been reported, and for its attached form, the size range of 230–400 nm and AMTD of 352 nm [103]. An AMTD of 1.28 nm for unattached $^{218}$Po was in the range of 1.04–1.55 nm, and that for unattached $^{214}$Pb of 1.30 nm was in the range of 1.10–1.60 nm, while the AMAD for attached $^{214}$Pb was 353 nm in the range of 240–550 nm and that for attached $^{214}$Bi was 380 in the range of 250–540 nm [104]. Zhang et al. [105] have demonstrated a slight size increase of unattached RnP with increasing aerosol particle concentration (e.g., AMD of $^{218}$Po increases from $0.94 \pm 0.17$ nm to $0.98 \pm 0.13$ nm when $N$ increases from 1.058 mm$^{-3}$ to 10.46 mm$^{-3}$). The typical values of those parameters have been adopted in lung dosimetry calculation by Marsh and Birchall [106].

**Table 2.** Activity median thermodynamic diameters (AMTDs) and their geometric standard deviations (GSD) at various aerosol concentrations ($N$) and concentrations of RnP and TnP potential $\alpha$-energy ($E_{\alpha RnP}$ and $E_{\alpha TnP}$), measured in radon chamber by Tu et al. [95].

| $N$ mm$^{-3}$ | $E_\alpha$ μJ m$^{-3}$ | | Unattached | | | | Attached | | | |
|---|---|---|---|---|---|---|---|---|---|---|
| | | | RnP | | TnP | | RnP | | TnP | |
| | $E_{\alpha RnP}$ | $E_{\alpha TnP}$ | AMTD | GSD | AMTD | GSD | AMTD | GSD | AMTD | GSD |
| 2.3 | 1.4 | 1.52 | 2 | 1.36 | 2 | 1.42 | 153 | 2.44 | 165 | 1.78 |
| 5.2 | 3.45 | 13.5 | 3 | 1.77 | 4 | 1.73 | 181 | 2.04 | 162 | 2.22 |
| 30 | 5.95 | 26.6 | 2 | 1.49 | 2 | 1.48 | 174 | 2.30 | 183 | 2.06 |
| 180 | 1.31 | 7.32 | 4 | 1.95 | 2 | 1.47 | 136 | 2.35 | 120 | 2.04 |

In theoretical calculations, various AMD values are considered for unattached RnP, including, for instance, 0.9 nm [58], 0.5–5 nm [107], or 0.5–1 nm [108]. Or, instead of attached–unattached, the particle size is classified as nucleation (50 nm), accumulation (250 nm), and coarse modes (1500 nm) [109].

While the attachment of RnP and TnP species to background aerosol particles is related to the aerosol concentration (Equation (1)), deposition and filtration of particles strongly depend on the particle size [56,66,110]. Therefore, the behaviour of the unattached and attached RnP and TnP is governed by the concentration of the background aerosol particles and their size distribution.

It has been shown [66,111,112] that when there is no particle source indoors, most indoor particles are of outdoor origin and brought in by air penetration. The indoor/outdoor particle concentration ratio may range from 0 [111] to 2.46 [112,113], and it increases with particle diameter, reaching the highest values between 100 nm and 400 nm [111]. The number particle concentration indoors results from competition between particle sources (their penetration from outdoor air and production by human activity indoors) and particle sinks (through ventilation, deposition, and filtration). This can be described as [66,110,114–117]

$$\frac{\mathrm{d}N}{\mathrm{d}t} = p_{\text{out}\to\text{in}}\lambda_{\text{v}} N^{\text{out}} - (\lambda_{\text{v}} + \lambda_{\text{d}} + \lambda_{\text{f}})N + \frac{Q_{\text{P}}}{V} \tag{22}$$

with:

$N$, $N^{\text{out}}$—indoor, outdoor particle number concentrations, $m^{-3}$, $cm^{-3}$ (particles per $m^3$, $cm^3$);
$p_{\text{out}\rightarrow\text{in}}$—outdoor $\rightarrow$ indoor particle penetration coefficient;
$\lambda_v$—ventilation rate constant, $s^{-1}$;
$\lambda_d$—deposition rate constant, $s^{-1}$;
$\lambda_f$—filtration rate constant, $s^{-1}$;
$Q_p$—particle generation rate of indoor sources, $s^{-1}$ (particles per s);
$V$—room volume, $m^3$, $cm^{-3}$.

Coagulation and condensation are not included in Equation (22). Because $p_{\text{out}\rightarrow\text{in}}$ [66,117], $\lambda_v$ [66,111,112], $\lambda_d$ [97,111], and $\lambda_f$ [110] depend on particle size, their values in the equation should be considered either as for a selected size or as averaged over all particle sizes.

The solution of Equation (22) under steady-state conditions [117],

$$N = \frac{p_{\text{out}\rightarrow\text{in}}\lambda_v N^{\text{out}}}{\lambda_v + \lambda_d + \lambda_f} + \frac{Q_p}{V(\lambda_v + \lambda_d + \lambda_f)}(1 - e^{-(\lambda_v + \lambda_d + \lambda_f)}), \tag{23}$$

shows the particle number concentration in indoor air as a result of their sources and sinks. Coagulation and condensation are not taken into consideration.

Before human activity starts to produce indoor particles at time $t = 0$ (and $Q_p = 0$), $N(0)$ is equal to the first term on the right side of Equation (23) and is considered the baseline or background level of $N$ before an event by human activity. With particle generation started, $\Delta N(t) = N(t) - N(0)$ begins to grow and reaches its maximum value $\Delta N(t_M)$ at time $t = t_M$ after the indoor particle source has been stopped. $\Delta N(t)$ decay follows. Thus, $\Delta N(t_M)$ shows the maximum increase in particle number concentration during an event caused by human activity. Assuming that $N(0)$ is constant during a short human activity event, Equation (23) can be used to calculate the particle generation rate of indoor sources [66,117]:

$$Q_p = \Delta N(t)\frac{V(\lambda_v + \lambda_d + \lambda_f)}{1 - e^{-(\lambda_v + \lambda_d + \lambda_f)t}}. \tag{24}$$

The result for $\Delta N(t_M)/2$ gives the average particle generation rate $Q_p$. If this $Q_p$ is multiplied by $t_M$ (duration of particle generation), the total number of particles emitted during an activated source is obtained.

The $(\lambda_v + \lambda_d + \lambda_f)$ sum can be obtained from the slope of the exponential decay of $N(t)$ with time, starting after $\Delta N_t(t_M)$, by using the relation [66,117]

$$\ln \Delta N(t) = \ln \Delta N(t_{M+}) - (\lambda_v + \lambda_d + \lambda_f)\,t,\, t > t_M, \tag{25}$$

in which $\Delta N(t_{M+})$ is a value at a point after the maximum, where the exponential form of the $\Delta N(t)$ curve decay appears.

As for the background aerosol, the activity concentrations of aerosol particles of the individual radon and thoron products in a room are also a result of the competition between their sources and sinks. Radon and thoron entry from outside air and sub-floor space and their exhalations from the building material (and the much lower extent of using water and gas) are their sources. Their sinks are, in addition to ventilation (aeration), deposition, and filtration, radioactive transformations ($\lambda_r$, r—radionuclide), and for unattached species, they also include attachment to the background aerosol particles ($\lambda_a$).

By applying the Jacobi room model under the steady-state conditions [54,56,57,61,103,118–121], the activity concentrations of individual radionuclide ($A$/Bq $m^{-3}$) can be calculated.

Equations for RnP and TnP are shown separately because of the specific characteristics of their chains. Table 3 shows the values of the physical quantities determining generations and losses of radon and thoron short-lived products to be used in the equations below, as they have been either measured or adopted as relevant values in modelling.

**Table 3.** Values of physical quantities determining generations and losses of (a) radon and (b) thoron short-lived products in applying the Jacobi room model: $Q_{Rn}$—radon generation rate of indoor sources, $N$—number concentration of aerosol particles, $\lambda_v$—ventilation rate constant, $\lambda_a$—attachment rate constant of the short-lived product to aerosol particles, $\beta$—attachment coefficient (cf. Equation (1)), $\lambda_d^u$—deposition rate constant of the unattached short-lived products, $\lambda_d^a$—deposition rate constant of the attached short-lived products, $v_d^u$—deposition velocity of the unattached short-lived products, $v_d^a$—deposition velocity of the attached short-lived products, $v_d^{eff}$—effective deposition velocity (Equation (3)), $r$—recoil fraction of short-lived products from aerosol particles. Where, in a row, three values are given for $\lambda_a$, $\lambda_d^u$, or $\lambda_d^a$, they refer to $^{218}$Po, $^{214}$Pb, and $^{214}$Bi, respectively.

| | | | | | **(a) Radon** | | | | | |
|---|---|---|---|---|---|---|---|---|---|---|
| $Q_{Rn}$ q m$^{-3}$ s$^{-1}$ | $N$ mm$^{-3}$ | $\lambda_v$ h$^{-1}$ | $\beta/10^{-3}$ cm$^3$ h$^{-1}$ | $\lambda_a$ h$^{-1}$ | $\lambda_d^u$ h$^{-1}$ | $\lambda_d^a$ h$^{-1}$ | $v_d^u$ m h$^{-1}$ | $v_d^a$ m h$^{-1}$ | $r$ | Reference |
| | 0.3−1.0 | | | 20−180 | 1−200 | 0.7 | | | | Jacobi [119] |
| | | 0.5−1.25 | | | | 1−200 | | | | Bruno [122] |
| | | <0.3 | 2.2−4.7 | | 10 | 0.1 | 2 | 0.2 | | Porstendörfer [56] |
| | | 1−2 | | | 30 | 0.3 | | | | Porstendörfer [56] |
| | | | | 86 | 100 | 0.1 | | | | Zarcone et al. [123] |
| | | 0.55 0.2−1.5 | | 50 5−500 | 20 10−40 | 0.2 0.1−0.4 | | | | Knutson [120] |
| 138 | 7−389 | <0.5 | 5.2 | | 54 | 0.21 | | | 0.83 | Reineking and Porstendörfer [54] |
| | | | | | | | | $^{218}$Po 0.72−1.4 | | Gadgil et al. [124] |
| | | 15 | | 50−200 | | | 8 | 0.08 | | Tu et al. [95] |
| | | | | | | | 4.5−9.3 | 4.5−9.3 | | Morawska and Jamriska [97] |
| | 100 | 0.57 2−81 | 3 | 300 | 46.8 30−67 | 0.47 0.33−0.67 | | | 0.80 | Islam et al. [125] |
| 7−14 | 2.9 1.3−4.6 | 0.3 | 91 56−184 | 170 94−354 | 0.225 0.05−0.66 | | | | 0.63 | El-Hussein [118] |
| | 260 | <0.3 | | 3600 | | | | | 0.47 0.28 | El-Hussein [118] |
| | | 0.2−0.25 | | 1.5−930 | 20 | 0.2 | | | 0.83 | Huet [79] |
| | | 0.59−0.6 | | 50−52 105−112 0.5−0.8 | 102−103 112−120 0.6−2.1 | 4.9−5.0 0.9−1.0 3.7−4.0 | | | | Nikolopoulos and Vogiannis [126] |

**Table 3.** *Cont.*

| | | | | (a) Radon | | | | | | |
|---|---|---|---|---|---|---|---|---|---|---|
| $Q_{Rn}$ q m⁻³ s⁻¹ | $N$ mm⁻³ | $\lambda_v$ h⁻¹ | $\beta/10^{-3}$ cm³ h⁻¹ | $\lambda_a$ h⁻¹ | $\lambda_d^u$ h⁻¹ | $\lambda_d^a$ h⁻¹ | $v_d^u$ m h⁻¹ | $v_d^a$ m h⁻¹ | $r$ | Reference |
| | | 0.59−0.6 | | 200−203<br>90−96<br>0.8−0.9 | 169−171<br>98−105<br>0.5−0.6 | 0.9−1.1<br>0.25−0.26<br>1.8−2.1 | | | | Nikolopoulos and Vogiannis [126] |
| | | 0.55<br>0.1−2 | | 50<br>10−100 | 20 | 0.2 | | | | Nikezić and Stevanović [109] |
| | | | | | 0.075 | | | | | Mishra et al. [127] |
| | | 0.1−1.0 | | | 3−110 | 0.015−0.35 | | | | Stevanovic et al. [58] |
| | 1−10 | | | 60−170<br>41−120<br>43−122 | 39−47<br>30−36<br>31−37 | (20−40) × 10⁻⁴<br>(7−1.5) × 10⁻⁴<br>(9−1.5) × 10⁻⁴ | | | | Stevanovic et al. [128] |
| | | 0.1−1.0 | | | 10−100 | 0.012−0.46 | | | | Stevanovic et al. [129] |
| | 0.6−50 | | | | | | ²¹²Pb: 0.28 | 0.11 | | Meisenberg and Tschiersch [61] |
| 5−11 | 2.6<br>1.3−4.3 | 0.4 | | 67<br>23−103 | 94<br>36−172 | 0.12<br>0.05−0.43 | | | | Mohery et al. [103] |
| 5−11 | 29<br>8−43 | 0.5 | | 69<br>24−108 | 110<br>28−202 | 0.09<br>0.05−0.42 | | | 0.54<br>0.24−1 | Mohery et al. [103] |
| | | 0.55 | | 50 | 20 | 0.2 | | | | Yu and Nikezic [130] |
| | | | | | | | 0.045 | | | Li et al. [131] |
| | | | | | | $v_d^{eff}$: 0.126 | | | | Rout et al. [57] |
| | 20 | 1 | | | | $v_d^{eff}$: 0.169 | | | | Mishra et al. [132] |

**Table 3.** *Cont.*

| (b) Thoron | | | | | | |
|---|---|---|---|---|---|---|
| $N$ $\mathbf{mm^{-3}}$ | $\lambda_\mathbf{v}$ $\mathbf{h^{-1}}$ | $\beta/10^{-3}$ $\mathbf{cm^3\,h^{-1}}$ | $\lambda_\mathrm{d}^\mathrm{u}$ $\mathbf{h^{-1}}$ | $v_\mathrm{d}^\mathrm{u}$ $\mathbf{m\,h^{-1}}$ | $v_\mathrm{d}^\mathrm{a}$ $\mathbf{m\,h^{-1}}$ | **Reference** |
| | | | | | $^{212}$Pb: 0.36−1.1 | Gadgil et al. [124] |
| | | | $0.132 \pm 0.004$ | | | Mishra et al. [127] |
| 30 | 0.5−1.0 | | | | 0.075 | Mishra et al. [127] |
| | | | | $v_\mathrm{d}^\mathrm{eff}$0.083 | 0.028 | Mishra et al. [59] |
| 0.6−50 | | | | $^{212}$Pb: 0.28 | 0.11 | Meisenberg and Tschiersch [61] |
| | | | | $v_\mathrm{d}^\mathrm{eff}$: 0.059 | | Rout et al. [57] |
| 20 | 1 | | | $v_\mathrm{d}^\mathrm{eff}$: 0.079 | | Mishra et al. [132] |

Radon gas:

$$A_{\text{Rn}} = \frac{Q_{\text{Rn}} + \lambda_{\text{v}} A_{\text{Rn}}^{\text{out}}}{\lambda_{\text{Rn}} + \lambda_{\text{v}}}, \tag{26}$$

with $Q_{\text{Rn}}$ (Bq m$^{-3}$ s$^{-1}$) the radon generation rate of indoor sources and $A_{\text{Rn}}$ and $A_{\text{Rn}}^{\text{out}}$ the radon activity concentration indoors and outdoors, respectively.

Unattached RnP

$$A_{\text{218Po}}^{\text{u}} = \frac{\lambda_{\text{218Po}} A_{\text{Rn}}}{\lambda_{\text{218Po}} + \lambda_{\text{a}} + \lambda_{\text{d}}^{\text{u}} + \lambda_{\text{v}} + \lambda_{\text{f}}} \tag{27}$$

$$A_{\text{214Pb}}^{\text{u}} = \frac{\lambda_{\text{214Pb}} A_{\text{218Po}}^{\text{u}} + r_{\text{218Po}} \lambda_{\text{214Pb}} A_{\text{218Po}}^{\text{a}}}{\lambda_{\text{214Pb}} + \lambda_{\text{a}} + \lambda_{\text{d}}^{\text{u}} + \lambda_{\text{v}} + \lambda_{\text{f}}} \tag{28}$$

$$A_{\text{214Bi}}^{\text{u}} = \frac{\lambda_{\text{214Bi}} A_{\text{214Pb}}^{\text{u}}}{\lambda_{\text{214Bi}} + \lambda_{\text{a}} + \lambda_{\text{d}}^{\text{u}} + \lambda_{\text{v}} + \lambda_{\text{f}}} \tag{29}$$

$$A_{\text{214Po}}^{\text{u}} = \frac{\lambda_{\text{214Po}} A_{\text{214Bi}}^{\text{u}}}{\lambda_{\text{214Po}} + \lambda_{\text{a}} + \lambda_{\text{d}}^{\text{u}} + \lambda_{\text{v}} + \lambda_{\text{f}}}, \tag{30}$$

and attached RnP

$$A_{\text{218Po}}^{\text{a}} = \frac{\lambda_{\text{v}} A_{\text{218Po}}^{\text{out,a}} + \lambda_{\text{a}} A_{\text{218Po}}^{\text{u}}}{\lambda_{\text{218Po}} + \lambda_{\text{d}}^{\text{a}} + \lambda_{\text{v}} + \lambda_{\text{f}}} \tag{31}$$

$$A_{\text{214Pb}}^{\text{a}} = \frac{\lambda_{\text{v}} A_{\text{214Pb}}^{\text{out,a}} + \lambda_{\text{a}} A_{\text{214Pb}}^{\text{u}} + (1 - r_{\text{218Po}}) \lambda_{\text{214Pb}} A_{\text{218Po}}^{\text{a}}}{\lambda_{\text{214Pb}} + \lambda_{\text{d}}^{\text{a}} + \lambda_{\text{v}} + \lambda_{\text{f}}} \tag{32}$$

$$A_{\text{214Bi}}^{\text{a}} = \frac{\lambda_{\text{v}} A_{\text{214Bi}}^{\text{out,a}} + \lambda_{\text{a}} A_{\text{214Bi}}^{\text{u}} + \lambda_{\text{214Bi}} A_{\text{214Pb}}^{\text{a}}}{\lambda_{\text{214Bi}} + \lambda_{\text{d}}^{\text{a}} + \lambda_{\text{v}} + \lambda_{\text{f}}} \tag{33}$$

$$A_{\text{214Po}}^{\text{a}} = \frac{\lambda_{\text{v}} A_{\text{214Po}}^{\text{out,a}} + \lambda_{\text{a}} A_{\text{214Po}}^{\text{u}} + \lambda_{\text{214Po}} A_{\text{214Bi}}^{\text{a}}}{\lambda_{\text{214Po}} + \lambda_{\text{d}}^{\text{a}} + \lambda_{\text{v}} + \lambda_{\text{f}}}. \tag{34}$$

First-term in Equations (31)$-$(34) (e.g., $\lambda_{\text{v}} A_{\text{218Po}}^{\text{out, a}}$) represents contributions of attached RnP in outdoor air entering the room through ventilation. This is justified because aerosol particles of a diameter between 100 nm and 400 nm have a high outdoor $\rightarrow$ indoor particle penetration coefficient $p_{\text{out}\rightarrow\text{in}}$ [111]. For the same reason, these contributions due to unattached RnP outdoors are not included in Equations (27)$-$(30). Perhaps this term would be more correctly expressed through $p_{\text{out}\rightarrow\text{in}}$ rather than through $\lambda_{\text{v}}$ (cf. Equation (23)). It is often considered negligible and omitted in model calculations [57,121].

Because of the big difference in their half-lives, radon and thoron behave differently indoors [60]. For instance, radon half-life is long enough to enable radon to be distributed uniformly in a closed room [60,133–135]. On the other hand, because of its short half-life, thoron activity concentration decreases exponentially with the distance from the exhalation surface. Provided only diffusion is considered, this decrease can be mathematically described as [9,134,136]

$$A_{\text{Tn}}(x) = A_{\text{Tn}}(0) e^{-\frac{x}{L_{\text{Tn}}}} \tag{35}$$

in which $A_{\text{Tn}}(0)$ and $A_{\text{Tn}}(x)$ stand for thoron activity concentration at the exhalation surface and a distance $x$ far from it, $\lambda_{\text{Tn}}$ is the rate constant of thoron $\alpha$-transformation, $D_{\text{Tn}}$ is its diffusion constant in air, and $L_{\text{Tn}} = \sqrt{D_{\text{Tn}}/\lambda_{\text{Tn}}}$ is its diffusion length in air. At a distance of $10-40$ cm from the surface, $A_{\text{Tn}}(x)$ reaches its asymptotic level, as measured in the middle of the room [134]. This should be borne in mind when designing the measurement points and reporting thoron levels and their doses. Nor may radon be considered uniformly distributed at a ventilation rate > 0.5 h$^{-1}$ [137].

While air movement in a closed room (e.g., using a fan) does not change radon activity concentration unless at ventilation rates > 0.5 h$^{-1}$ [137], it increases thoron activity

concentration by distributing thoron uniformly in the room, thus establishing its average level $\overline{A_{\mathrm{Tn}}}$ [138,139]. In a room at a ventilation rate constant $\lambda_{\mathrm{v}}$ of volume $V$ and thoron exhalation surface $S$ (e.g., floor or wall), $\overline{A_{\mathrm{Tn}}}$ can be expressed by its exhalation rate from indoor surfaces ($Q_{\mathrm{Tn}}/\mathrm{Bq\ m^{-2}\ s^{-1}}$) as [61]:

$$\overline{A_{\mathrm{Tn}}} = \frac{Q_{\mathrm{Tn}}S/V + \lambda_{\mathrm{v}}A_{\mathrm{Tn}}^{\mathrm{out}}}{(\lambda_{\mathrm{Tn}} + \lambda_{\mathrm{v}})}. \tag{36}$$

From the dosimetry point of view, of all TnPs, only $^{212}$Pb and $^{212}$Bi are interesting (as evident in Equation (7)). Because of the very short half-life of Tn, the contribution from the outdoor thoron ($A_{\mathrm{Tn}}^{\mathrm{out}}$) to $\overline{A_{\mathrm{Tn}}}$ is generally negligible unless at very high ventilation rates and may be omitted. Considering their half-lives, $^{216}$Po can be distributed homogeneously in a room, while thoron cannot. $^{216}$Po is close to the secular equilibrium with thoron, and one may write [60]:

$$A_{^{216}\mathrm{Po}} \approx \overline{A_{\mathrm{Tn}}} \tag{37}$$

$$A_{^{212}\mathrm{Pb}}^{\mathrm{u}} = \frac{\lambda_{^{212}\mathrm{Pb}}A_{^{216}\mathrm{Po}}}{\lambda_{^{212}\mathrm{Pb}} + \lambda_{\mathrm{a}} + \lambda_{\mathrm{d}}^{\mathrm{u}} + \lambda_{\mathrm{v}} + \lambda_{\mathrm{f}}} \tag{38}$$

$$A_{^{212}\mathrm{Bi}}^{\mathrm{u}} = \frac{\lambda_{^{212}\mathrm{Bi}}A_{^{212}\mathrm{Pb}}^{\mathrm{u}}}{\lambda_{^{212}\mathrm{Bi}} + \lambda_{\mathrm{a}} + \lambda_{\mathrm{d}}^{\mathrm{u}} + \lambda_{\mathrm{v}} + \lambda_{\mathrm{f}}} \tag{39}$$

$$A_{^{212}\mathrm{Pb}}^{\mathrm{a}} = \frac{\lambda_{\mathrm{a}}A_{^{212}\mathrm{Pb}}^{\mathrm{u}}}{\lambda_{^{212}\mathrm{Pb}} + \lambda_{\mathrm{d}}^{\mathrm{a}} + \lambda_{\mathrm{v}} + \lambda_{\mathrm{f}}} \tag{40}$$

$$A_{^{212}\mathrm{Bi}}^{\mathrm{a}} = \frac{\lambda_{\mathrm{a}}A_{^{212}\mathrm{Bi}}^{\mathrm{u}}}{\lambda_{^{212}\mathrm{Bi}} + \lambda_{\mathrm{d}}^{\mathrm{a}} + \lambda_{\mathrm{v}} + \lambda_{\mathrm{f}}}. \tag{41}$$

## 4. Radon and Thoron Sources

### 4.1. Ground

Radon and thoron sources are $^{238}$U and $^{232}$Th, respectively (Figure 1), in the ground on which a building stands, in the building material, and in used tap water and gas. Because of their half-lives and associated diffusion lengths, radon enters a building predominantly from the ground, while for thoron, its exhalation from the floor and walls in a room is generally its predominant source indoors. Therefore, the correlation between radon and thoron activity concentrations indoors is weak [38], if any exists [46]. Nonetheless, the opposite situation also appears.

As reported by Kemski et al. [140], based on 4019 results for Germany, the radon concentration in soil gas can range from kBq m$^{-3}$ to MBq m$^{-3}$ levels, being highest in the igneous and lowest in sedimentary geological units. Surprisingly, soil radon influences indoor levels only above its threshold of 20 kBq m$^{-3}$ in soil gas [140,141], and below that, there is no correlation between the radon levels in soil gas and indoor air [142,143]. According to 3512 data from 11 geological units in Lombardy, Italy [144], the highest indoor radon levels of around 200 Bq m$^{-3}$ geometric mean appeared over dolomite rocks, acid rocks (igneous, metamorphic, granite, gneiss), and debris (landslides, rock falls). In a Polish study in 129 buildings, the highest indoor radon levels (reaching up to 400 Bq m$^{-3}$) were measured in the Sudetes, with crystalline igneous and metamorphic rock on the surface [145]. In 800 buildings surveyed in Switzerland, the highest geometric means were observed in the Southern Alps, characterised by crystalline rocks of various chemical compositions [146]. In Belgium, about 10,000 houses were surveyed, and the results showed the highest indoor radon levels on sandstone, quartz-phyllite, psammite, and greywacke; moderate levels on phyllite, mica-sandstone, limestone, chalk, and calc-shale; and lowest levels on sand, clay and marl [147]. Elevated indoor levels of radon and thoron are also expected in buildings on volcanic ground and made of volcanic material, as seen from the studies in Italy and Spain [148–150]. Also, in Hungary, the highest indoor radon activity concentrations in 6154 one-storey, no-basement houses were found on Cenozoic volcanic

rocks, followed by Paleozoic granites, with the lowest in houses constructed on sedimentary fluvial and partly Aeolian formations [151].

Tectonic and geological faults cause anomalies in the spatial distribution of radon in the soil, mainly expressed as increased radon activity concentration in soil gas [152,153]. Therefore, fault zones present potential radon-prone areas. In a study of 149 houses located in fossil regions, fault zones, and normal areas in the Aizawl district, India [154], the effect of faults on the indoor radon and thoron activity concentrations was observed. While GM (geometric mean) values of radon activity concentrations were highest within fault zones (48.5 Bq m$^{-3}$) and lowest in fossil regions (35 Bq m$^{-3}$), those of thoron were highest in normal areas (13.9 Bq m$^{-3}$) and lowest within fault zones (11.2 Bq m$^{-3}$), although single highest values for radon were in the normal area and those of thoron were in fault zones. Nonetheless, the differences were not significant.

In their radon measurements in 15 houses in the proximity of the Amer fault in La Vall d'en Bas in Spain, Moreno et al. [149] observed a tendency of increased indoor radon levels as they approached the fault: it was 20−30 Bq m$^{-3}$ at a distance of 2 km away and increased to 60−130 Bq m$^{-3}$ at a distance of fewer than 300 m to the fault.

Dai et al. [155] evaluated the efficacy of housing and geological characteristics to predict radon risk in DeKalb County, Georgia, USA. In the fault zones, indoor radon levels were more likely to exceed the USA action level (148 Bq m$^{-3}$); they were significantly positively correlated to gamma readings but significantly negatively related to the presence of a crawlspace foundation and its combination with a slab.

Elevated indoor radon levels may also be found over carbonates [156–158]. Generally, carbonates have a low content of uranium/radium, but the presence of the terra rossa soil (possibly enriched in uranium and with high radon emanation) and karstic phenomena (cracks and fissures) facilitate radon migration and thus enhance its entry into a house [158–160]. Peake [157] reported for Wisconsin, USA even higher values in 212 homes over carbonates (167 ± 174 Bq m$^{-3}$) than in 119 homes over granite (155 ± 181 Bq m$^{-3}$).

Based on 330 indoor air radon concentrations measured in kindergartens and schools in Slovenia [161], the highest average values were found in buildings on limestone and dolomite located in the karstic southwest part of the country, and also those covered by several tectonic faults [162]. This was ascribed to higher $^{226}$Ra content in carbonates (range: 12−270 Bq kg$^{-1}$, GM = 66 Bq kg$^{-1}$) in a study showing its distribution at 70 sites in Slovenia [163]. As further explained, $^{238}$U and $^{226}$Ra migration by water and the high permeability of carbonate led to a wash-out of more mobile uranium and increased the $^{226}$Ra/$^{238}$U ratio, for which a range of 0.8−3.2 and an arithmetic mean of two were observed. The radon emanation fraction of carbonates was also high (range: 0.010−0.55, GM = 0.18), second only to that of sea and lake sediments (range: 0.26−0.42, GM = 0.34) (Figure 3). Although correlations between radon emanation and uranium [164] and $^{226}$Ra content in soil are weak [162], a good correlation was observed between radium content and radon activity concentration in both soil gas and indoor air (Figure 4a,b) [163], thus predicting high indoor radon levels in karst buildings.

### 4.2. Building Material

Exhalation of radon and thoron from building materials can be their significant source indoors. Exhalation rates differ markedly from material to material. They have been systematically measured by Pillai et al. [165] in various building materials, as shown by their results in Table 4.

The results of Singh et al. [166] in Table 5 show how different exhalation rates from different building materials are reflected in different radon and thoron levels indoors. They attribute high radon levels in mud houses to direct radon exhalation from the floor and poor ventilation. Nonetheless, because of standard deviations of 37−47 Bq m$^{-3}$ for $A_{Rn}$, 15−19 Bq m$^{-3}$ for $A_{RnP}$, 43−57 Bq m$^{-3}$ for $A_{Tn}$, and 1.25−1.44 Bq m$^{-3}$ for $A_{TnP}$, differences between different materials do not seem significant.

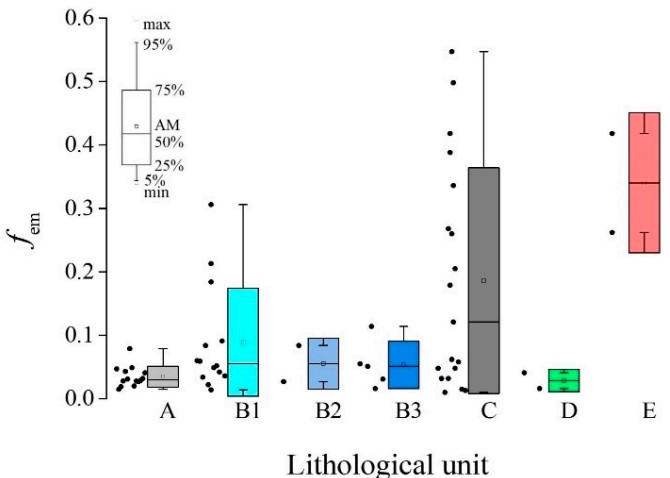

**Figure 3.** Box and whiskers diagram of the distribution of radon emanation fraction ($f_{em}$) concerning the lithological units at 58 points in Slovenia: A—alluvial and glacial deposits, B1—clastic sediments containing clay, B2—coarse clastic sediments, B3—flysch, C—carbonates, D—metamorphic rocks, E—sea and lake sediments. Adapted by Kardos et al. [162].

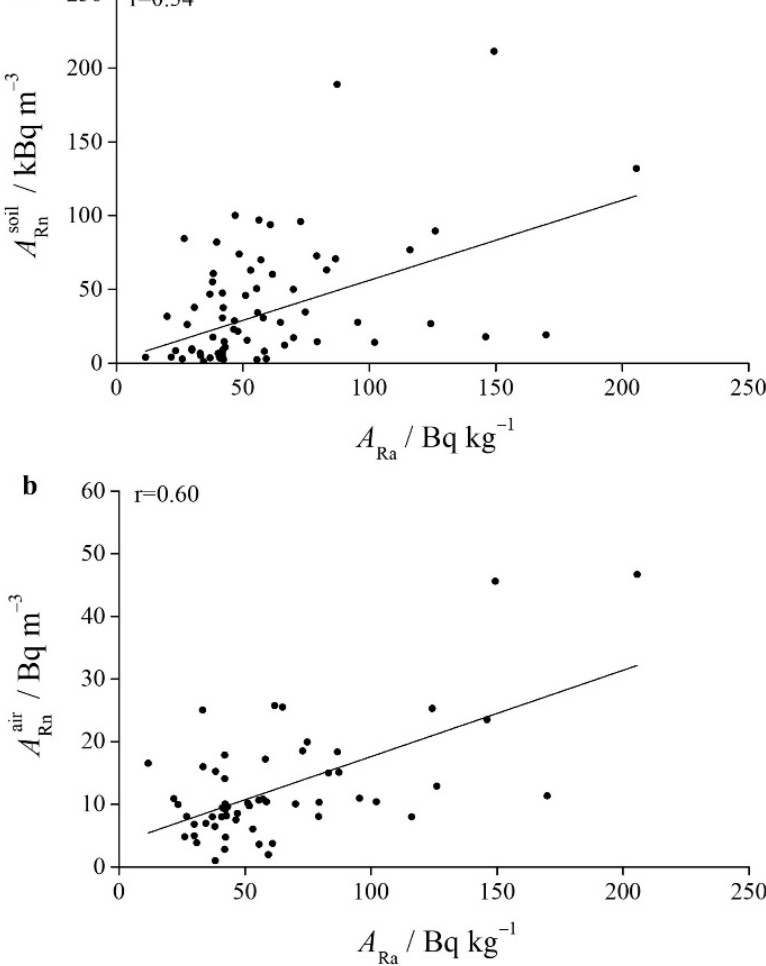

**Figure 4.** Correlation between $^{226}$Ra activity concentration in soil $A_{Ra}$ at 70 points in Slovenia and (**a**) radon activity concentration in soil gas ($A_{Rn}^{soil}$) and (**b**) radon activity concentration in outdoor air ($A_{Rn}^{air}$). Adapted by Kovács et al. [163].

**Table 4.** Ranges and geometric (GM) or arithmetic (AM *) means of radon exhalation rates (in mass unit mBq kg$^{-1}$ h$^{-1}$ and surface unit mBq m$^{-2}$ h$^{-1}$) of selected building materials by Pillai et al. [165].

| | | Radon Exhalation Rate | | | |
| | | mBq kg$^{-1}$ h$^{-1}$ | | mBq m$^{-2}$ h$^{-1}$ | |
| Material | No. of Samples | Range | GM or AM | Range | GM or AM |
|---|---|---|---|---|---|
| Sedimentary rock | 14 | 1.8−29 | 6.2×/:2.28 | 13−215 | 46.2×/:2.29 |
| Igneous rock | 9 | 13−156 | 31.5×/:2.24 | 94−1127 | 388×/:2.13 |
| River sand | 5 | 16−90 | 36.6×/:2.84 | 118−49 | 265×/:1.18 |
| Bricks | 10 | 4.2−76 | 18.6×/:2.14 | 31−551 | 135×/:2.14 |
| Cement | 11 | 24−41 | 31.7 ± 4.9 * | 170−297 | 229 ± 36 * |

**Table 5.** Geometric means (GMs) and ranges (in brackets) of indoor activity concentrations (Bq m$^{-3}$) of radon ($A_{\text{Rn}}$), thoron ($A_{\text{Tn}}$), and their short-lived products ($A_{\text{RnP}}$, $A_{\text{TnP}}$) in dwellings made of different building materials by Singh et al. [166].

| Building Material | No | $A_{\text{Rn}}$ | $A_{\text{RnP}}$ | $A_{\text{Tn}}$ | $A_{\text{TnP}}$ |
|---|---|---|---|---|---|
| Mud | 17 | 111 (45−180) | 37 (13−76) | 76 (14−151) | 2.69 (0.99−2.30) |
| Stone, cement plaster | 17 | 97 (42−208) | 38 (21−72) | 76 (9−196) | 2.66 (1.25−5.88) |
| Cement | 18 | 94 (41−200) | 31 (7.9−65) | 87 (10−253) | 2.26 (1.25−5.43) |

While a similarly weak dependence of indoor radon and thoron levels on the type of building material has also been observed in some other studies [46,154], on the other hand, there are reports on well-pronounced differences among building materials [50,167–171]. As an example, results obtained in 23 dwellings in Devon and Cornwall, United Kingdom, are pointed out [169]. The highest values of both $A_{\text{RnP}}$ and $A_{\text{TnP}}$ were obtained with granite (26 Bq m$^{-3}$, 0.68 Bq m$^{-3}$) and limestone (26 Bq m$^{-3}$, 0.77 Bq m$^{-3}$) as building materials.

Another example is the results obtained in 62 houses in Yamuna, Tons, and Kedar valleys in Garhwal, Himalaya, India [50]. Old-style mud houses are made of local mud and stone, with a roof covered by slates. When cement, stone, and brick are used for walls, for the floor, and for the roof, cement, concrete, and iron bars are used. For traditional wooden houses, mainly wood is used. Average activity concentrations of radon, thoron, and their products are higher on the ground floor than on the first floor, indicating that the ground is the prevailing source of radon and thoron. From the ground to the first floor (with cement, stone, and brick as building materials), $A_{\text{Rn}}$ decreased to 32% and $A_{\text{Tn}}$ to 58%, suggesting that building material is for thoron an important source.

The data on the building materials in the above tables are not satisfactory for critically evaluating the role of building materials in determining indoor RnP and TnP levels. For this purpose, more detailed information would be needed, including, for instance, on the nature and quality of the building material, size, and dimensions of floors and walls, aeration, occupancy, and living habits.

The trend of decreasing radon activity concentration from the basement (or ground floor) towards higher floors has been observed generally [28,146,160,170,172–176]. Nevertheless, there are places where this trend is not strictly followed, as obtained in 487 dwellings in Alto Lazio and 255 dwellings in Rome, Italy [150]. Although in Alto Lazio, the radon level decreases towards higher floors, this decrease is slow, i.e., on the first–second floor, its value is still 47% of the value in the basement for tuff and 59% for brick, thus indicating building material as an important radon source in addition to the ground. These percentages are similar in Rome, but here, the radon level on the ground floor is more than twice as high as that in the basement, thus further confirming building material as a strong radon source. In addition, in 14 buildings under study in Alto Lazio, Sciocchetti et al. [150] found 7 buildings in which concentrations of potential α-energy of RnP and TnP were higher on

the upper floors than in the basement. Measurements in 601 dwellings in Kuwait revealed the highest radon level in the basement, followed by the first floor, while it was the lowest on the ground floor [177].

Also very interesting are the results of radon and thoron measurements by Tu et al. [178] in 40 homes and six public buildings in New York, New Jersey, Colorado, Illinois, and Pennsylvania, USA. Their results (as concentrations of $\alpha$-potential energy $E_{\alpha RnP}$ and $E_{\alpha TnP}$) are divided into three groups with respect to $E_{\alpha RnP}$ levels as follows: group A, >400 nJ m$^{-3}$, group B, $100-400$ nJ m$^{-3}$, and group C, <100 nJ m$^{-3}$. In group B, the ratio between the ground floor and the basement values was 1.04 for $E_{\alpha RnP}$ and 2.4 for $E_{\alpha TnP}$. The correlation coefficient of the $E_{\alpha RnP}$ versus $E_{\alpha TnP}$ relationship was 0.78 for the basement and 0.16 for the ground floor. This suggests a common source of radon and thoron in the basement and separate sources on the ground floor. In the same study [178], in basements (B) and on ground floors (GF) in 16 buildings in New Jersey and 7 buildings in New York, the correlation coefficient for $E_{\alpha TnP}$ between the basement and ground floor was 0.99, and it was 0.79 for $E_{\alpha RnP}$. In seven buildings (30%), the GF/B ratio for thoron was above 1 (i.e., 1.29, 1.42, 2.25, 1.35, 1.6, 4.46, 1.21), and in two additional buildings, it was close to 1. These high values were accompanied by GF/B ratios for radon close to 1 (but always <1). In seven buildings with GF/B < 0.25 for thoron, GF/B for radon was also low at $0.04-0.38$. This suggests that in these buildings, the major thoron source is not the ground.

Nevertheless, caution is suggested when reading or reporting differences in radon and thoron levels on the ground and first floors. In rooms on the first floor of detached houses, which are usually less frequently entered and aerated than those on the ground floor, radon levels may be easily higher than downstairs [177].

### 4.3. Age of Building

The age of a building also affects its indoor radon and thoron levels. With ageing, failures and cracks in the main concrete slab and walls connecting the ground may occur, and radon entry from the ground is facilitated [179]. More important are changes in construction with time, such as in the type of building, use of building materials, hydro and thermal insulation (also doors and windows), heating systems, and many other details. According to a survey of 450 dwellings in South Korea [168], higher thoron levels were observed in older, detached, one-family houses (both of traditional and modern building material), but no change was observed in blocks of flats. A radon survey in 400 dwellings in Slovenia has shown lower average activity concentrations in houses built after 1966 (due to the steadily decreasing use of stone as a building material) but with exceptions of very high values ascribed to lower aeration rates in tight buildings with effective thermal insulation [180]. On the other hand, a thorough analysis by Kemski et al. [140] has shown a steady decrease in indoor radon levels in newer houses, but this is dependent on the region because of local specifics in construction. Nevertheless, they have strongly emphasised that because the indoor radon level depends on a great number of influencing parameters, considerable caution is required when discussing geology, building materials, and building ages without knowing all of the necessary information, including details. Similarly, Finne et al. [181] observed a considerable reduction in radon activity concentrations in newly built houses in Norway after the implementation of the new building regulations in 2010. The concentrations vary between different dwelling categories (e.g., in detached houses, the average radon concentration dropped from 76 to 40 Bq m$^{-3}$).

Here, it is also important to mention an enhanced energy retrofit of buildings by achieving better tightness. One of the very effective measures is the replacement of old windows, which can significantly increase indoor radon concentrations. Pampuri et al. [182] surveyed 154 buildings before and after energy remediation and revealed an increase in indoor radon concentration on average of 22% (increase in 100 buildings, decrease in 52 buildings, no change in 2 buildings). Collignan and Powaga [183] pointed out that any thermal retrofit process in the building must be associated with the relevant ventilation system to avoid a significant increase in indoor radon concentration. Towards greater

energy conservation in buildings, a Spanish study [184] revealed significantly lower radon concentrations in dwellings built in the traditional style than in new houses.

## 5. Temporal Variations in Radon and Thoron Levels in Indoor Air

Radon emanation from mineral grains, its migration in the ground, and thus its entry into the building depend on meteorological conditions. In addition, both radon and thoron (and thus RnP and TnP) levels indoors and their hourly, diurnal, and seasonal variations are also greatly influenced by the construction characteristics and the inhabitants' habits and indoor activities.

Two-day radon measurements in a weakly ventilated experimental house have shown that outdoor air pressure and humidity and water content in soil under the building had more of an effect on the indoor radon levels than the air temperature, rainfall, wind velocity, and soil temperature [185].

In another work [160], radon and meteorological parameters (indoor and outdoor air temperature and relative humidity, outdoor air pressure, rainfall height, and wind speed) were monitored year-long (frequency once an hour) in two living rooms in a high-radon dwelling in the Slovenian Karst, with one on the ground floor under normal household activities and the other on the first floor, which was closed and very rarely entered. The correlation coefficients (r) obtained between radon level and meteorological parameters based on the monthly averages of measured data showed the following general conclusions: (i) correlation coefficients (r) vary substantially (both in size and sign) between rooms and between months, (ii) on the ground floor, outdoor air temperature better correlates with the radon level than indoor temperature, and the opposite is valid for the first floor, (iii) a similar situation was seen for air humidity, (iv) on the ground floor, the correlation for outdoor air temperature was generally negative, while that for outdoor air humidity was positive, (v) ignoring months with $|r| \leq 0.1$, correlations for pressure were negative except for several months, (vi) no general regularity was observed between r values and values of the influential parameter except for indoor and outdoor air humidity on the ground floor and indoor and outdoor air temperature on the first floor, with higher r values being associated with higher parameter values, and (vii) large variations in correlation are attributed to the combined action of a great number of environmental parameters (often interrelated) [186,187] and residents' living habits. The interrelation between wind speed and air pressure [188] has not been evaluated.

Correlation coefficients were also calculated from the hourly values of parameters for the entire year [160]. The correlation between the radon concentration and the outdoor air temperature was moderately negative on both floors, thus identifying this parameter as the most influential [189]. The correlation coefficient was also moderate for indoor air temperature on the first floor and the temperature difference between indoors and outdoors on the ground floor. The outdoor relative air humidity had a poorer correlation with the radon level than the outdoor air temperature, which was negative on the ground floor and positive on the first floor. The indoor relative air humidity had a stronger effect on the ground than on the first floor. The correlation for pressure was minor and, on both floors, negative. The effect of wind appeared to be not negligible on the first floor, as observed in public buildings in Italy [190], and that of rain was very small in both rooms. Groves-Kirkby et al. [191] have critically evaluated the influence of meteorological parameters on radon behaviour indoors.

In the living room on the ground floor of the two-storey dwelling in the above study [160], typical diurnal variations of radon activity concentration were observed. Radon maxima coincided well with $T_{\mathrm{in}} - T_{\mathrm{out}}$ maxima (air temperature difference between indoors and outdoors). A thermal air lift, caused by $T_{\mathrm{in}} - T_{\mathrm{out}}$ increase, enhanced the room aeration by allowing the inflow of fresh air, but on the other hand, it also increased the entry of sub-slab radon-rich air into the room [192–195].

Seasonal variation in radon activity concentrations is generally reported by values in the spring, summer, autumn (or rainy season in tropics), and winter, and only seldom by monthly

radon activity concentrations [186,190,196–200]. It can often be approximated by a sinusoidal function, with a maximum in winter and a minimum in summer [159,170,186,198,201,202].

The winter-to-summer (W/S) ratio of the radon activity concentrations may differ substantially from country to country [170,196,202–205], from region to region within a country [198,199,206,207], from building to building in the same region [187], and from floor to floor in the same building [160,190,208]. It can be higher at higher altitudes (mountain versus valley), as observed in rooms on the first floor but not on the ground floor [170].

In 4742 dwellings in 21 regions in Italy [206], W/S values between 0.1 and 7.1 were found, with <1 value appearing in each region. In North Macedonia [170], the W/S ratio in 437 dwellings in eight regions ranged from 0.05 to 3.27, with an average of 0.47. In the UK [198], it ranged from 0.4 to 5.3 in 728 determinations, being <1 in 4% cases, >1.25 in 68% cases, and 0.75−1.25 in 27% cases. Substantially lower [149] or higher W/S values [159,160] could be observed in buildings connected to underground cavities or caves in karst regions.

Reports on the temporal variation of thoron activity concentration are modest in comparison to that of radon. In addition, reports on seasonal variations of activity concentrations of radon and thoron short-lived products have appeared only recently [45,49,50,166,209].

The diurnal variation of thoron activity concentration often follows that of radon, as in living rooms in two high-radon dwellings in Niška Banja, Serbia [210] (Figure 5). A significant increase in radon activity concentration overnight was observed only in house 2 (factor of about 1.5) but not in house 5 (factor of about 1.2). This is not always the case, and the two variations may also differ, as reported by Németh et al. [135].

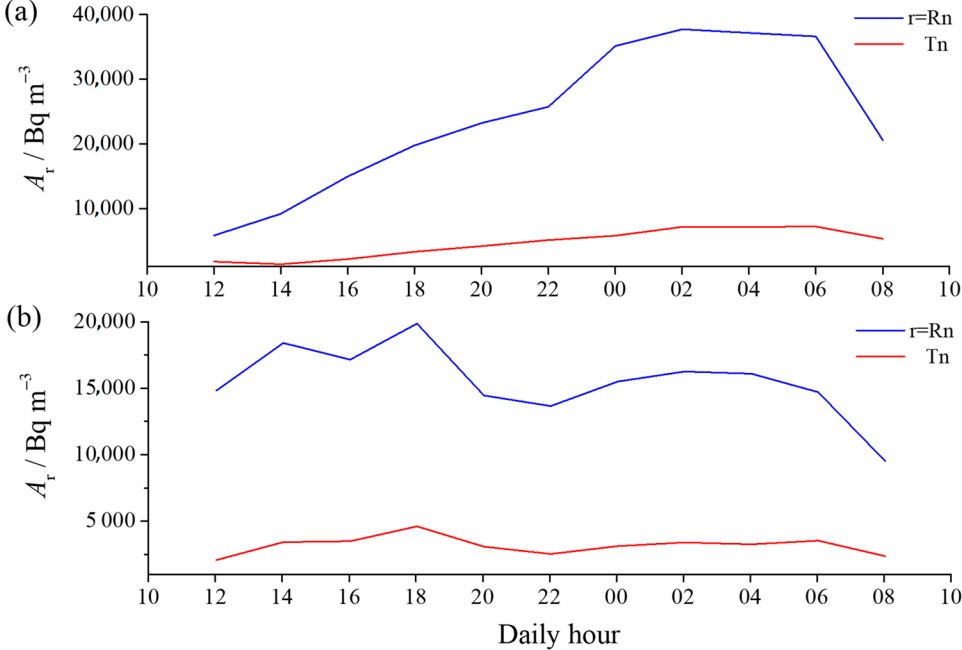

**Figure 5.** Diurnal variation of radon ($A_{\mathrm{Rn}}$) and thoron ($A_{\mathrm{Tn}}$) activity concentrations in the bedroom of (**a**) dwelling 2 and (**b**) dwelling 5 in Niška Banja, Serbia. Adapted by Vaupotič et al. [210].

In their paper, Ramola et al. [50] report on activity concentrations of radon, thoron, and their short-lived products (in attached and unattached forms), equilibrium factor, and unattached fraction obtained in the summer, winter, spring, and rainy season in 122 houses in Yamuna, Tons, and Kedar valleys in Garhwal Himalaya, India. Their average values for the summer and winter (for comparison with the radon data above) are summarised in Table 6. For $A_{\mathrm{Rn}}$, the W/S value was comparable to values obtained in Europe despite a difference in the climate. Only for $F_{\mathrm{Rn}}$ and $f^{\mathrm{u}}_{\mathrm{RnP}}$ were W/S values close to 1, and they ranged from 1.5 to 2 for other radon parameters. Except for $A^{\mathrm{a}}_{\mathrm{TnP}}$ and $F_{\mathrm{Rn}}$, W/S values for other thoron parameters were markedly lower than those for the related radon parameters.

**Table 6.** Average values of the parameters given in the first column, obtained in winter and summer, and the winter-to-summer ratio (W/S), in 122 houses in Yamuna, Tons, and Kedar valleys in Garhwal Himalaya, India by Ramola et al. [50].

| | Winter | | | Summer | | | |
| Parameter | Min | Max | GM | Min | Max | GM | W/S |
|---|---|---|---|---|---|---|---|
| $A_{Rn}$/Bq m$^{-3}$ | $36 \pm 4$ | $182 \pm 9$ | 66 | $5 \pm 1$ | $174 \pm 9$ | 34 | 1.94 |
| $A_{RnP}$/Bq m$^{-3}$ | $6.7 \pm 1$ | $65.1 \pm 2.5$ | 22.6 | $2.1 \pm 0.4$ | $37.6 \pm 1.9$ | 13.0 | 1.74 |
| $A_{RnP}^{a}$/Bq m$^{-3}$ | $5.7 \pm 1.1$ | 47.5 | 16.7 | $1.4 \pm 0.5$ | $34 \pm 2.7$ | 11.1 | 1.50 |
| $A_{RnP}^{u}$/Bq m$^{-3}$ | 0.2 | 27.8 | 2.6 | 0.26 | 13.0 | 1.3 | 2.0 |
| $f_{RnP}^{u}$ | 0.01 | 0.84 | 0.11 | 0.01 | 0.87 | 0.11 | 1.0 |
| $F_{Rn}$ | 0.10 | 0.91 | 0.34 | 0.10 | 0.83 | 0.37 | 0.92 |
| $A_{Tn}$/Bq m$^{-3}$ | $2 \pm 1$ | $210 \pm 10$ | 33 | $4 \pm 1$ | $195 \pm 10$ | 27 | 1.22 |
| $A_{TnP}$/Bq m$^{-3}$ | $0.3 \pm 0.1$ | $4.8 \pm 0.2$ | 1.6 | $0.3 \pm 0.1$ | $11.2 \pm 0.3$ | 1.2 | 1.33 |
| $A_{TnP}^{a}$/Bq m$^{-3}$ | $0.3 \pm 0.1$ | $4.7 \pm 0.3$ | 1.4 | $0.2 \pm 0.1$ | $9.6 \pm 0.5$ | 0.9 | 1.56 |
| $A_{TnP}^{u}$/Bq m$^{-3}$ | <0.05 | 1.8 | 0.1 | 0.1 | 1.8 | 0.3 | 0.33 |
| $f_{TnP}^{u}$ | 0.01 | 0.46 | 0.07 | 0.01 | 0.70 | 0.21 | 0.33 |
| $F_{Tn}$ | 0.01 | 0.21 | 0.05 | 0.01 | 0.31 | 0.05 | 1.0 |

Activity concentrations of radon, thoron, and their short-lived products in wintertime, together with their W/S ratio, obtained in India, are also reported for 96 dwellings in Hamirpur district, Himachal Pradesh [211], 35 dwellings in Pauri Garhwal, Uttarakhand, Himalaya [45], 27 dwellings in Yamuna and Tons valleys, Garhwal, Himalaya [49], and 25 dwellings in Rajpur region, Uttarakhand, Himalaya [209]. The values are similar to those in Table 6 for the related parameters. In total, W/S values in the Himalaya area occur in the ranges of $1.49-2.24$ for $A_{Rn}$, $1.22-2.44$ for $A_{RnP}$, $1.28-5.25$ for $A_{Tn}$, and $1.26-1.88$ for $A_{TnP}$.

Another study reported year-round indoor radon, thoron, and their short-lived product concentrations (together with particulate matter ($PM_{2.5}$)) in Beijing and Changchun, China and Aomori, Japan, three metropolises with different air quality levels. The annual mean equilibrium equivalent radon (EERC) and thoron (EETC) concentrations were 17.2 and 1.1 Bq m$^{-3}$ in Beijing, 19.4 and 1.3 Bq m$^{-3}$ in Changchun, and 10.8 and 0.9 Bq m$^{-3}$ in Aomori, respectively, being the highest in winter [212].

To successfully implement sustainable policies for indoor air quality, it is crucial to consider the temporal variations in radon levels caused by meteorological parameters.

## 6. Impact of Human Activities on Radon Behaviour Indoors

Because of its adverse effects on health, we intend to keep the activity concentration of radon and thoron and their short-lived products indoors below reasonably low levels. For this purpose, the first concern is how to minimise radon and thoron sources in a building. Entry of the radioactive gas from the ground can be efficiently reduced or even stopped by constructing high-quality floor slabs and walls contacting the ground, which is particularly necessary for radon-prone areas (geology, lithology). In addition, building materials should be carefully selected, bearing in mind uranium and thorium content. Only seldom can using water and natural gas in a household increase radon levels significantly [6,7].

When a building is already constructed, the indoor levels of radon, thoron, and their short-lived products are strongly influenced by human behaviour in addition to meteorological parameters. Human habits and activities play an important role in sustainable indoor air quality management.

### 6.1. Ventilation

Ventilation, either mechanically or through opening windows, is simple and has proven to be a dominant human-influenced factor in controlling indoor radon levels [118,123,131,213–217], and it is even more effective than using a building mate-

rial low in uranium and thorium [214]. However, cost-effectiveness should be borne in mind, as more fresh air requires enhanced heating in the wintertime [218]. An example of reducing activity concentrations by increasing the ventilation rate from $0.2\,\text{h}^{-1}$ to $0.5\,\text{h}^{-1}$ in a dwelling made of clay and brick is presented for radon, thoron, and their products in Table 7 [215]. In this small $\lambda_v$ range, the slope of the relationship is linear, but it turns into an exponential trend afterwards [119,124,214]. In the last row (i.e., 0.2/0.5) in Table 7a,b, the ratio of the values at $0.2\,\text{h}^{-1}$ and $0.5\,\text{h}^{-1}$ flow rates is shown. The ratio is similar for Rn and $^{218}$Po and Tn and $^{216}$Po, being higher for the former two. It is higher for $^{212}$Pb, $^{212}$Bi, and $^{212}$Po than for $^{214}$Pb, $^{214}$Bi, and $^{214}$Po.

**Table 7.** Effect of the ventilation rate constant ($\lambda_v/\text{h}^{-1}$) in a dwelling made of clay and brick on the activity concentrations ($A_r/\text{Bq}\,\text{m}^{-3}$, r—radionuclide) of (a) radon and its products and (b) thoron and its products (* in the last row indicated as '0.2/0.5'; the ratio of the values at $\lambda_v = 0.20\,\text{h}^{-1}$ and $\lambda_v = 0.50\,\text{h}^{-1}$ is shown) by Misdaq et al. [215].

**(a) Radon and its products**

| $\lambda_v$ | $A_{\text{Rn}}$ | $A_{218\text{Po}}$ | $A_{214\text{Pb}}$ | $A_{214\text{Bi}}$ | $A_{214\text{Po}}$ | $F_{\text{Rn}}$ |
|---|---|---|---|---|---|---|
| 0.20 | $69.4 \pm 4.9$ | $54.8 \pm 3.0$ | $42.2 \pm 1.9$ | $34.6 \pm 1.7$ | $34.3 \pm 2.0$ | $0.58 \pm 0.03$ |
| 0.25 | $67.7 \pm 4.7$ | $53.4 \pm 2.1$ | $40.1 \pm 2.2$ | $32.1 \pm 1.7$ | $31.7 \pm 1.6$ | $0.56 \pm 0.03$ |
| 0.30 | $58.4 \pm 4.0$ | $46.1 \pm 3.2$ | $33.7 \pm 2.4$ | $26.3 \pm 1.3$ | $26.0 \pm 1.8$ | $0.55 \pm 0.03$ |
| 0.35 | $49.4 \pm 3.4$ | $39.0 \pm 1.9$ | $28.1 \pm 1.4$ | $21.9 \pm 1.1$ | $21.7 \pm 0.7$ | $0.54 \pm 0.03$ |
| 0.50 | $38.4 \pm 2.7$ | $29.9 \pm 1.5$ | $19.8 \pm 1.0$ | $14.4 \pm 0.9$ | $14.7 \pm 0.7$ | $0.48 \pm 0.02$ |
| 0.2/0.5 * | 1.81 | 1.83 | 2.13 | 2.40 | 2.33 | 1.21 |

**(b) Thoron and its products**

| $\lambda_v$ | $A_{\text{Tn}}$ | $A_{216\text{Po}}$ | $A_{212\text{Pb}}$ | $A_{212\text{Bi}}$ | $A_{212\text{Po}}$ | $F_{\text{Tn}}$ |
|---|---|---|---|---|---|---|
| 0.20 | $2.22 \pm 0.11$ | $2.19 \pm 0.12$ | $0.260 \pm 0.010$ | $0.158 \pm 0.008$ | $0.156 \pm 0.006$ | $0.110 \pm 0.008$ |
| 0.25 | $2.51 \pm 0.14$ | $2.48 \pm 0.11$ | $0.273 \pm 0.010$ | $0.155 \pm 0.008$ | $0.154 \pm 0.007$ | $0.100 \pm 0.005$ |
| 0.30 | $2.75 \pm 0.13$ | $2.72 \pm 0.10$ | $0.270 \pm 0.010$ | $0.147 \pm 0.008$ | $0.145 \pm 0.007$ | $0.090 \pm 0.004$ |
| 0.35 | $2.30 \pm 0.11$ | $2.27 \pm 0.13$ | $0.210 \pm 0.010$ | $0.108 \pm 0.006$ | $0.107 \pm 0.008$ | $0.080 \pm 0.004$ |
| 0.50 | $1.48 \pm 0.10$ | $1.46 \pm 0.07$ | $0.100 \pm 0.005$ | $0.048 \pm 0.002$ | $0.047 \pm 0.003$ | $0.060 \pm 0.004$ |
| 0.2/0.5 * | 1.50 | 1.50 | 2.60 | 3.29 | 3.32 | 1.83 |

Recently, some authors have included simulations of radon and carbon dioxide concentrations in their studies of ventilation efficiency and compared them to measurements. Concerning radon, García-Tobar [219] proposed a methodology for estimating radon levels in a naturally and mechanically ventilated dwelling in a radon-prone area using the CONTAM program. Further, García-Tobar [220] analysed the influence of weather on indoor radon concentration in a new multi-storey building in a radon-prone area. Dovjak et al. [221] checked legislative requirements and recommendations for ventilation efficiency. For a renovated school, with the average measured radon concentration of $200-1000\,\text{Bq}\,\text{m}^{-3}$, radon concentration was simulated by varying the design ventilation rates (DVRs). The DVRs were insufficient in 24% of cases, according to the EU, and in 56% of cases, according to the WHO guidelines. In the following study, Dovjak et al. [222] checked the ventilation efficiency of radon and carbon dioxide concentrations with measurements and simulations in a small apartment. The results indicate the need for further research on outdoor/indoor interactions, emphasizing ventilation in the built environment.

### 6.2. Air Conditioning

Radon levels are also affected by air conditioning [30,177,223,224]. As an example, Table 8 presents results obtained when running the air-conditioning system under different conditions in the auditorium of the Lublin University of Technology, Lublin, Poland [223]. Comparing rows 1 and 2 reveals that switching off the system increased $A_{\text{Rn}}$, $A_{\text{RnP}}$, and $N$. Averages of $A_{\text{Rn}}$ and $A_{\text{RnP}}$ were lower in the daytime than overnight, but no difference in $N$ (rows 3 and 4) was discerned. Increasing the flow rate from $5400\,\text{m}^3\,\text{h}^{-1}$ to $7200\,\text{m}^3\,\text{h}^{-1}$

(rows 5 and 6) reduced $A_{RnP}$ but did not change either $A_{Rn}$ or $N$; only a further flow rate increase to 9000 m$^3$ h$^{-1}$ (rows 6 and 7) reduced both $A_{Rn}$ and $N$, leaving $A_{RnP}$ unchanged. From additional measurements in the same auditorium, the influence of air conditioning on the partitioning between unattached and attached radon products was obtained [224]. During the 'on' periods, radon concentration was decreased (due to the addition of fresh outside air), as was the attached fraction of radon products, while the contribution of the unattached radon products was increased. While no change in particle number concentration was observed (except its range), the mass concentration of PM1 particles (smaller than 1 µm) decreased from 45.7 µg cm$^{-3}$ to 21.7 µg cm$^{-3}$. Presumably, bigger and heavier particles were deposited and replaced by smaller and lighter ones. Interestingly, a good correlation (r = 0.61) was observed for the $A_{RnP}^a$ versus $N$ relationship during the 'off' periods, but no correlation was observed during the 'on' periods.

**Table 8.** Activity concentrations of radon ($A_{Rn}$) and its short-lived products ($A_{RnP}$) and number concentration of aerosol ($N$) when air conditioning was in operation under various regimes in the auditorium at the Lublin University of Technology, Lublin, Poland, by Grządziel et al. [223].

|   | Operation Time | Airflow m$^3$ h$^{-1}$ | Addition of Fresh Air % | $A_{Rn}$ Bq m$^{-3}$ | $A_{RnP}$ Bq m$^{-3}$ | $N$ mm$^{-3}$ |
|---|---|---|---|---|---|---|
| 1 | all time | | | 25 ± 16 | 1.2 ± 0.9 | 4.1 ± 3.6 |
| 2 | 8 a.m.–8 p.m. | | | 31 ± 19 | 5.0 ± 3.1 | 5.2 ± 4.1 |
| 3 | 6 a.m.–8 p.m. | | | 30 ± 18 | 4.4 ± 3.1 | 5.2 ± 4.6 |
| 4 | 9 p.m.–5 a.m. | | | 33 ± 19 | 6.1 ± 2.8 | 5.2 ± 3.1 |
| 5 | all time | 5400 | 85 | 17 ± 8 | 2.5 ± 1.1 | 2.7 ± 2.6 |
| 6 | all time | 7200 | 85 | 20 ± 10 | 0.8 ± 0.7 | 2.5 ± 1.7 |
| 7 | all time | 9000 | 85 | 12 ± 6 | 0.9 ± 0.6 | 1.8 ± 1.1 |

An increase in radon levels overnight, when the air conditioning was switched off, was also observed in two offices, one on the 6th floor of an 8-storey building and the other on the 10th floor of a 44-storey building [30]. Their results in November were, for the 6th floor, from 9 am to 5 pm, $A_{Rn}$ = 34 Bq m$^{-3}$, $A_{RnP}$ = 12 Bq m$^{-3}$; for the whole day, $A_{Rn}$ = 58 Bq m$^{-3}$, $A_{RnP}$ = 28 Bq m$^{-3}$; for the 10th floor, from 9 am to 5 pm, $A_{Rn}$ = 13 Bq m$^{-3}$, $A_{RnP}$ = 5 Bq m$^{-3}$; for the whole day, $A_{Rn}$ = 19 Bq m$^{-3}$, $A_{RnP}$ = 8 Bq m$^{-3}$. The reason for this increase at such high floors could not result from soil gas or outdoor air; therefore, the authors ascribed it to building materials and an airtight structure. It has also been shown that a central air-conditioning system (in which conditioned air circulates through a number of rooms) for a building ensures lower radon levels than separate windows or room devices (in which conditioned air circulates in a closed loop of the room) [177].

### 6.3. Air Filtration

Air filtration reduces indoor levels of radon short-lived products efficiently, but not of radon [110,139,225–229]. This efficiency was checked in the 24.3 m$^3$ radon chamber at the National Institute of Radiological Sciences (NIRS), Chiba, Japan [225], where an AMU-04 (Airtech, Tokyo, Japan) air cleaner was used with a HEPA (high-efficiency particulate air) filter. They compared $A_{Rn}$, $A_{RnP}$, and $f_{RnP}^u$ for periods when the cleaner was in operation (on: $A_{Rn}$ = 11,000 Bq m$^{-3}$, $A_{RnP}$ = 180 Bq m$^{-3}$, and $f_{RnP}^u$ = 0.65) and when it was not in use (off: $A_{Rn}$ = 11,000 Bq m$^{-3}$, $A_{RnP}$ = 860 Bq m$^{-3}$, and $f_{RnP}^u$ = 0.12). During filtration, the concentration of aerosol markedly decreased but without any shift in particle size distribution. With the unchanged radon level, the activity concentration of its products was substantially reduced, and the fraction of unattached products substantially increased, thus leading to an increase in the radon dose conversion factor (cf. Equations (14)−(16)). Air filtration using a HEPA filter alone and in combination with a carbon filter in the kitchen and shower room in a flat on the fifth floor in Kobe, Japan showed [228] a similar efficiency for radon products removal but also a slight, though statistically significant, decrease in radon concentration.

A study carried out in the HMGU thoron experimental house at the Helmholtz Zentrum München, Germany [139] demonstrated how the concentrations of potential $\alpha$-energy of the attached radon and thoron short-lived products ($E^a_{\alpha RnP}$, $E^a_{\alpha TnP}$) decreased concomitantly with a decrease in aerosol concentration during air filtration using a HEPA filter at various radon and thoron levels, aerosol concentrations, and ventilation rates. On the other hand, both $E^u_{\alpha RnP}$ and $E^u_{\alpha TnP}$ increased steadily, and in 84 h, when aerosol concentration decreased from 2000 cm$^{-3}$ to 600 cm$^{-3}$, the former reached 120% and the latter 400% of its initial value. The authors supported this behaviour with theoretical calculations.

The following filtration experiments were carried out during weekends in a playroom of a kindergarten in Ljubljana, Slovenia [110]. For this purpose, a 125 W mobile air cleaner was run at airflow rates (m$^3$ h$^{-1}$) at 300 in step 1, 700 in step 2, and 1200 in step 3. Activity concentrations of radon and its short-lived products in the unattached and attached form were measured (once every two hours) using an EQF3020-2 device (Sarad, Dresden, Germany). The number concentration and size distribution of aerosol particles in the 5–530 nm size range were measured with an SMPS + C instrument, Series 5.400 with the medium DMA unit (Grimm, Hamburg, Germany). The instrument gives (every four minutes) the total number concentration of particles ($N$), particle number concentrations in each of the 44 size windows ($N_d$), particle number size distribution (d$N$/dln$d$, with $d$ being the electrical mobility-equivalent particle diameter), and the geometric mean of particle diameter ($d_{GM}$).

When the cleaner was turned on, the total number concentration ($N$) of particles, with size distribution maximised at 100 nm [110], started to decrease and reached its minimum in about an hour due to the removal of the >30 nm particles. After that, the concentration of >30 nm particles ($N_{30-200}$) continued to decrease slowly towards the end of filtration, while the concentration of smaller particles ($N_{<10}$) started to increase, reaching its maximum in the middle of step 2 and remaining constant afterwards. They were ascribed to the operation of the electric motor in the cleaner [230]. These changes in aerosol characteristics influenced the behaviour of radon products as follows: $A_{RnP}$ and $F_{Rn}$ decreased and $f^u_{RnP}$ increased rapidly in the beginning and more slowly afterwards. According to Equation (16), $f^u_{RnP}$ = 0.09 prior to filtration would give $f_{DC}$ for nasal breathing of 7.7 mSv WLM$^{-1}$, and $f^u_{RnP}$ = 0.48 at the end of filtration, 14.3 mSv WLM$^{-1}$, i.e., about twice as much. On the other hand, during this time, $A_{RnP}$ decreased from 1200 Bq m$^{-3}$ to 200 Bq m$^{-3}$ by a factor of six. Thus, at the end of air filtration under the above conditions, the effective dose rate was reduced by a factor of about three, which is more than reported earlier [228].

### 6.4. Emission of Nanoparticles

Concentration and size distribution of nanoparticles in indoor air determine the degree of secular equilibrium between radon and its products and thoron and its products ($F_{Rn}$, $F_{Tn}$) and also the fraction of unattached radon and thoron products ($f^u_{RnP}$, $f^u_{TnP}$), two key parameters in radon and thoron dosimetry. Therefore, knowledge of the sources of particles and their behaviour indoors is a prerequisite to reliably assess exposure to radon and thoron and to understand its dependence on the habits and activities of inhabitants in addition to the environmental parameters.

In periods without any human activity, aerosol concentration indoors is usually between 0.7 mm$^{-3}$ and 19 mm$^{-3}$ [113,230,231], and, in most cases, it is lower than outdoors [66,112,113]. Any human activity generates aerosol particles, even simple walking [232]. A large variety of human activities indoors comprises simple entering and leaving rooms, preparation of meals (following various recipes and using different household appliances), personal affairs (smoking, showering, drying hair, using the sauna), cleaning (with different tools), laundry, ventilation (naturally by opening windows or mechanically with fans and air conditioners), heating (using electricity or different fuels, either centrally for the building or locally in rooms), and others, depending on the habits of occupants. According to a review of over 20 indoor activities by He et al. [115] and Morawska et al. [233], the highest number concentration of aerosol and the highest particle

emission rate were measured when using the stove, grill, and fan heater, and the lowest were during hair drying and dusting. Some of these results are shown in Table 9 [115].

**Table 9.** Median values and standard deviations (SD) of the maximum number concentration of aerosol particles ($N_M$) emitted during an indoor activity, the ratio between $N_M$ and background ($N_b$) aerosol concentration ($N_M/N_b$), and the particle generation–emission rate ($Q_p$), measured in houses in the residential suburb of Brisbane, Australia by He et al. [115].

| | $N_M/\text{mm}^{-3}$ | | $N_M/N_b$ | | $Q_p/10^{11}\,\text{s}^{-1}$ | |
| --- | --- | --- | --- | --- | --- | --- |
| **Activity** | **Median** | **SD** | **Median** | **SD** | **Median** | **SD** |
| Cooking | 126 | 177 | 10.3 | 19.3 | 5.67 | 8.61 |
| Frying | 154 | 21.3 | 10.0 | 6.1 | 4.75 | 2.34 |
| Grilling | 161 | 69.9 | 8.69 | 5.27 | 7.34 | 5.06 |
| Microwave | 16.3 | 28.6 | 1.12 | 1.55 | 0.55 | 1.94 |
| Stove | 179 | 287 | 12.5 | 10.5 | 7.33 | 51.4 |
| Toasting | 114 | 160 | 6.34 | 7.44 | 6.75 | 16.7 |
| Smoking | 26.6 | 13.6 | 1.54 | 0.96 | 1.91 | 1.92 |
| Vacuuming | 41.3 | 17.6 | 1.51 | 1.17 | 0.97 | 1.57 |
| Sweep floor | 34.9 | 5.86 | 1.05 | 0.01 | 0.12 | 0.02 |
| Washing | 30.9 | 18.5 | 1.30 | 0.83 | 0.96 | 2.60 |
| Dusting | 14.1 | | 1.00 | | | |
| Fan heater | 87.1 | | 27.2 | | 4.07 | |
| Hair dryer | 9.5 | | 1.06 | | 0.11 | |
| Shower | 10.7 | | 1.37 | | 0.78 | |
| Washing machine | 11.1 | | 1.18 | | 0.15 | |

Often, the highest particle concentration was exhibited by burning candles (not included in the above review); for instance, 241 $\text{mm}^{-3}$ [234], 400–500 $\text{mm}^{-3}$ [230], and 1200 $\text{mm}^{-3}$ [235]. In contrast, candle burning was found by Hussein et al. [111] to be the weakest particle source in comparison to smoking, frying, and using a stove and aroma lamp.

In order to provide a rough insight into the sizes of particles emitted during different indoor activities, Table 10 shows part of the results obtained in a townhouse in Reston, VA, USA [236]. As is evident, except for the citronella candle, 2/3 of particles emitted indoors fall into the 10–100 nm size range (% in second column).

**Table 10.** Examples of particle number concentration ($N/\text{mm}^{-3}$) in selected ranges of particle diameter emitted during various indoor activities in a townhouse in Reston, VA, USA (with % for the contribution of 10–100 nm particles) by Wallace [236].

| | **Ranges of Particle Diameter/nm** | | | | |
| --- | --- | --- | --- | --- | --- |
| **Particle Source** | **10–100 (%)** | **100–200** | **200–450** | **450–950** | **Total** |
| No source | 2.56 (75) | 0.68 | 0.183 | 0.018 | 3.38 |
| Outdoors | 9.52 (31) | 18.35 | 3.03 | 0.016 | 30.92 |
| Tea | 5.76 (99) | 0.058 | | | 5.82 |
| Tea + toast | 9.53 (100) | 0.001 | | | 9.53 |
| Breakfast | 19.97 (99) | 0.117 | | 0.013 | 20.10 |
| Fried eggs | 22.51 (90) | 2.20 | 0.317 | 0.053 | 25.08 |
| Dinner | 30.46 (92) | 2.31 | 0.321 | 0.041 | 33.13 |
| Tortillas | 39.70 (81) | 8.39 | 0.823 | 0.072 | 48.99 |
| Broiled fish | 47.23 (95) | 2.63 | 0.098 | 0.017 | 49.98 |
| Gas oven | 29.74 (95) | 1.40 | 0.068 | 0.021 | 31.23 |
| Incense | 6.68 (69) | 2.31 | 0.722 | | 9.71 |
| Citronella candle | 3.14 (45) | 1.61 | 1.63 | 0.576 | 6.96 |

Vargas Trassierra et al. [63] have shown how differences in aerosol characteristics formed during smoking traditional and electronic cigarettes exhibited differences in con-

centration of the potential $\alpha$-energy of both unattached and attached radon short-lived products ($E^{\mathrm{u}}_{\alpha\mathrm{RnP}}$ and $E^{\mathrm{a}}_{\alpha\mathrm{RnP}}$), as well as their equilibrium factor ($F_{\mathrm{Rn}}$). Experiments were carried out in the radon chamber at the National Institute of Ionizing Radiation Metrology (INMRI-ENEA), Rome, Italy. Only part of their results is presented in Table 11. As smoking started, the concentration of aerosol particles started to increase and, as a consequence [56,101,176], the contribution of the attached radon products and equilibrium factors started to be enhanced and that of unattached products was reduced. These processes did not stop when smoking was finished in 12 min but continued because the creation and formation of radon product species need time [126] to reach a maximum in $E^{\mathrm{a}}_{\alpha\mathrm{RnP}}$ and $F_{\mathrm{Rn}}$ and a minimum in $E^{\mathrm{u}}_{\alpha\mathrm{RnP}}$, with a delay of several hours. Although the increase in aerosol concentration with a traditional cigarette was substantially higher than with electronic ones, the increase in $E^{\mathrm{a}}_{\alpha\mathrm{RnP}}$ and $F_{\mathrm{Rn}}$ was higher with the electronic cigarette (last row in Table 11).

**Table 11.** Average values of number aerosol concentration ($N/\mathrm{mm}^{-3}$), the geometric mean of particle diameter ($d_{\mathrm{GM}}/\mathrm{nm}$), the concentration of potential $\alpha$-energy of attached ($E^{\mathrm{a}}_{\alpha\mathrm{RnP}}/\mathrm{MeV\ cm}^{-3}$) and unattached radon short-lived products ($E^{\mathrm{u}}_{\alpha\mathrm{RnP}}/\mathrm{MeV\ cm}^{-3}$), and their equilibrium factor ($F_{\mathrm{Rn}}$), as measured before and at the end of 12 min of 'smoking' traditional and electronic cigarettes in the radon chamber (at 1.6 kBq m$^{-3}$ radon concentration) at INMRI-ENEA, Rome, Italy; the last row shows the ratio between the maximum and background values by Vargas Trassierra et al. [63].

| | Traditional Cigarette | | | | | Electronic Cigarette | | | | |
|---|---|---|---|---|---|---|---|---|---|---|
| **Smoking Step** | $N$ | $d_{\mathrm{GM}}$ | $E^{\mathrm{a}}_{\alpha\mathrm{RnP}}$ | $E^{\mathrm{u}}_{\alpha\mathrm{RnP}}$ | $F_{\mathrm{Rn}}$ | $N$ | $d_{\mathrm{GM}}$ | $E^{\mathrm{a}}_{\alpha\mathrm{RnP}}$ | $E^{\mathrm{u}}_{\alpha\mathrm{RnP}}$ | $F_{\mathrm{Rn}}$ |
| a: before | 3.36 | | 14.1 | 2.14 | 0.31 | 3.36 | 67 | 7.47 | 3.18 | 0.23 |
| b: at the end | 506 | | 18.6 | 0.48 | 0.38 | 62.9 | 87 | 12.6 | 2.53 | 0.30 |
| b/a | 150 | | 1.32 | 0.22 | 1.22 | 18 | | 1.69 | 0.80 | 1.30 |

Another experiment [237] shows the effect of the aerosol particles from smoking an electronic cigarette on the particle size distribution in indoor air and air in a radon chamber. Activity size distributions are clearly bimodal, with modes corresponding to the unattached RnP and RnP attached to the smoke particles of the electronic cigarette. The activity of the unattached RnP occurs at about 1 nm in diameter size and is nearly monodisperse. Activity size distribution of the attached RnP, occurring in a size range between 0.1 and 0.4 µm diameter, is heterodisperse and corresponds to the distribution of the electronic cigarette smoke particles.

### 6.5. Cigarette Smoking and Candle Burning—Detailed Description

The role of the aerosol particles generated during cigarette smoking and candle burning on the behaviour of radon short-lived products in indoor air has been presented in more detail by results obtained in a basement kitchen of a family house in a suburban area in Ljubljana city, Slovenia [231,235]. For this purpose, activity concentrations of radon and its short-lived products in the unattached and attached form were measured using an EQF3020-2 (and seldom EQF3220) device (Sarad, Dresden, Germany), and the number concentration and size distribution of aerosol particles in the 5–530 nm size range were monitored with an SMPS + C instrument, Series 5.400, with the medium DMA unit (Grimm, Hamburg, Germany), as described in Section 6.3 Air Filtration.

### 6.5.1. Cigarette Smoking

Several experiments were carried out, and the results of one are presented in Figure 6. The initial aerosol concentration $N$ prior to smoking was 3.6 mm$^{-3}$ (Figure 6a), showing a bimodal particle size distribution, with diameters from 5 to 9 nm and from 10 to 200 nm (Figure 7), resulting in an overall geometric mean of particle diameter $d_{\mathrm{GM}} = 44.5$ nm (Figure 6a). As a person started to smoke a cigarette (at 10:24), $N$ started to increase and reached its maximum of 435 mm$^{-3}$ at the end of smoking (at 10:40). This value was

lower than the 504 mm$^{-3}$ recently reported by Vargas Trassierra et al. [63], similar to that reported by Reineking et al. [230], and much higher than values reported by other authors, which are between 240 mm$^{-3}$ and 300 mm$^{-3}$ [101,118,234] or even lower [111,115]. The highest contribution to the maximum *N* originated from particles with a diameter around 100 nm, followed by those with diameters around 70 nm and 150 nm, and substantially lower than others, as evidenced by Figure 8a, showing the number concentration of particles in the following size windows: 9.3 nm (8.5−9.3 nm), 29.6 nm (27.1−29.6 nm), 68.4 nm (62.2−68.4 nm), 101.4 nm (91.8−101.4 nm), 153.9 nm (138.3−153.9 nm), 215.2 nm (192−215.2 nm), and 308.7 nm (272.9−308.7 nm). The initial number fraction of particles smaller than 10 nm ($x_{<10}$) was reduced from 0.14 to less than 0.001 (Figure 6b). After its maximum, *N* started to decrease, but even after 4 h, it did not fall to its initial value. Changes in particle size distribution are shown in Figure 7 at four times during the experiment: (i) prior to smoking (8:40), (ii) at $d_{GM}$ = minimum (10:26), (iii) at *N* = maximum (10:51), and (iv) two hours after smoking (12:39). The simultaneous decrease in *N* and increase in $d_{GM}$ have been attributed to coagulation of the emitted particles [234].

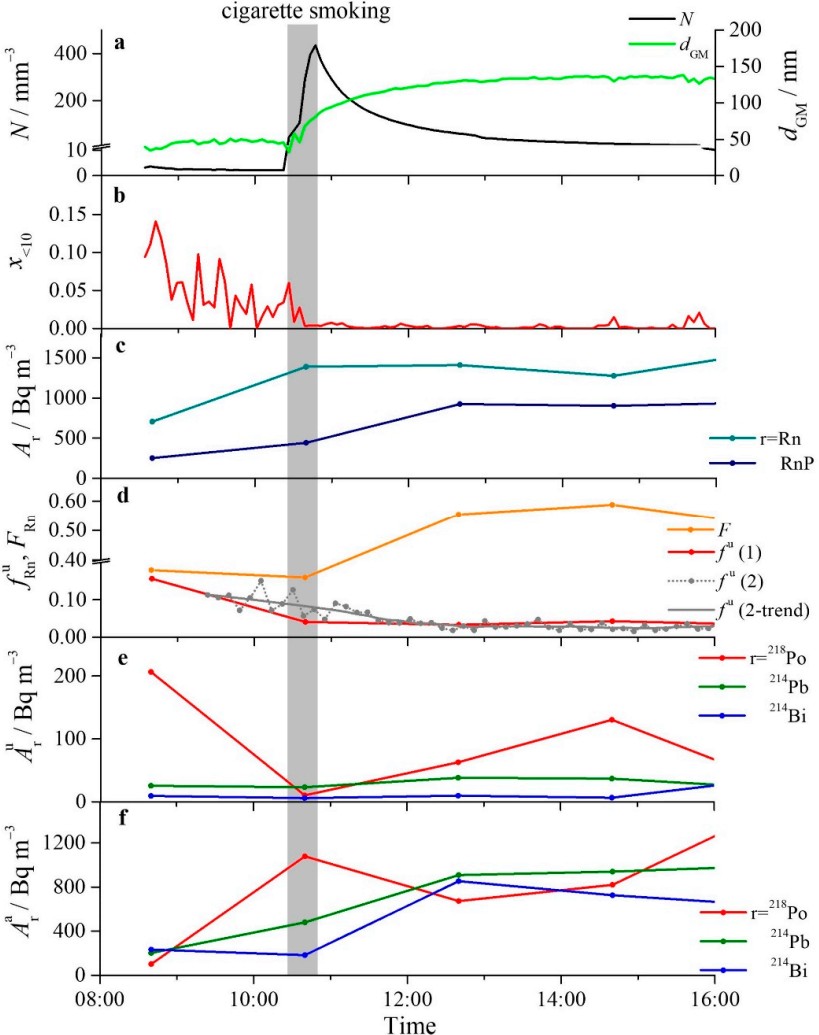

**Figure 6.** Cigarette smoking experiment with a time run of (**a**) the total number concentration of aerosol particles (*N*) and the geometric mean of their diameters ($d_{GM}$), (**b**) the number fraction of particles smaller than 10 nm ($x_{<10}$), (**c**) the activity concentration of radon ($A_{Rn}$) and radon products ($A_{RnP}$), (**d**) the unattached fraction of radon short-lived products ($f_{Rn}^{u}$) and equilibrium factor between radon and its short-lived products ($F_{Rn}$), (**e**) activity concentrations of the unattached RnP species, $^{218}$Po ($A_{218Po}^{u}$), $^{214}$Pb ($A_{214Pb}^{u}$), and $^{214}$Bi ($A_{214Bi}^{u}$), and (**f**) activity concentrations of the attached RnP species, $^{218}$Po ($A_{218Po}^{a}$), $^{214}$Pb ($A_{214Pb}^{a}$), and $^{214}$Bi ($A_{214Bi}^{a}$).

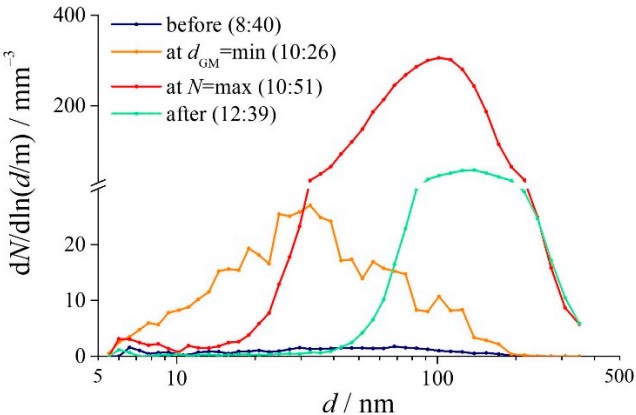

**Figure 7.** Cigarette smoking experiment: number size distribution of aerosol nanoparticles before cigarette smoking (at 8:40), at $d_{GM}$ = minimum (at 10:26), at $N$ = maximum (at 10:51), and after smoking (at 12:39).

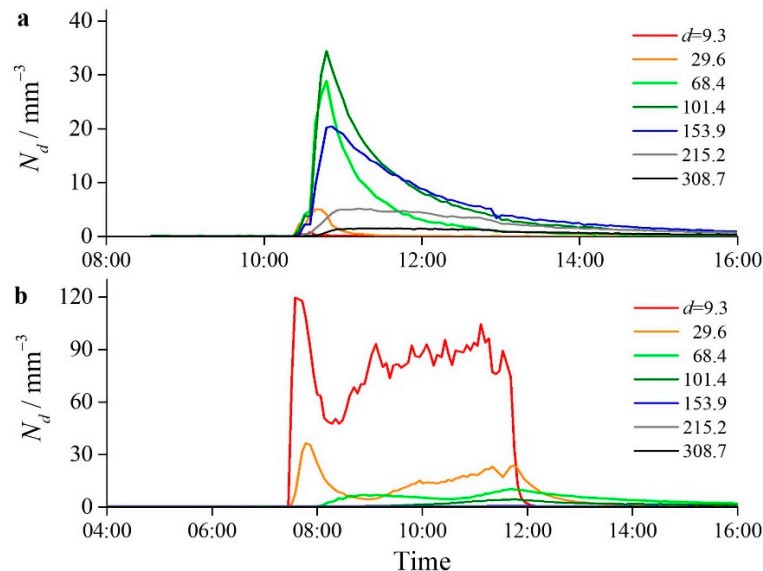

**Figure 8.** Number concentrations ($N_d$) of particles in the selected size windows of 9.3, 29.6, 68.4, 101.4, 153.9, 215.2, and 308.7 nm during (**a**) cigarette smoking and (**b**) candle burning.

An increase in $N$ was expected to cause an enhancement in $F_{Rn}$ and a reduction of $f_{RnP}^u$ [56,101,176], and this is also seen in Figures 6d and 6c, respectively; the former changed from 0.36 to 0.60, and the latter from 0.16 to 0.03. The decrease in $f_{RnP}^u$ resulted from a marked decrease in the contribution of the unattached $^{218}$Po species (Figure 6e) and an increase in the attached $^{218}$Po and $^{214}$Bi species (Figure 6f), bearing in mind that the coefficient at $^{218}$Po in Equation (4) is only 0.11 and that at $^{214}$Bi is 0.38. For $N = 3.6$ mm$^{-3}$, Equation (17) would predict $f_{RnP}^u$ =0.11, which is close to the measured value of 0.16 prior to smoking, and for 434.9 mm$^{-3}$, the prediction is 0.00092, which is far below our measured value of 0.04 during the smoking. Towards the next RnP measurement at 12:40, $f_{RnP}^u$ did not change significantly despite the $N$ decrease to 60 mm$^{-3}$. This may not be understood as a violation of the relationship expressed by Equation (17) but instead ascribed to the nature of the processes monitored. Changes in the number concentration and size distribution of aerosol particles during smoking were rapid, lasting only minutes. It is not expected that a change in size distribution will cause an immediate redistribution between unattached and attached RnP. It is more likely to influence only the newly born RnP atoms and clusters. The creation of RnP atoms through radioactive transformations takes time; also, their neutralisation, clustering, and attachment to and detachment from aerosol particles through

recoil are processes with defined values of rate constants. Taking this into account, the calculation would show [126] that a time delay of even more than an hour [139] necessarily appears between a change in aerosol characteristics and a change in $f_{RnP}^u$. Therefore, for short changes in aerosol, the above relationship appears to be masked or even totally obscured [107,231,235].

### 6.5.2. Candle Burning

For these experiments, ordinary tea candles were used, which burned down in three to four hours, and the maximum number concentration of particles emitted reached values from 1300 mm$^{-3}$ to 1600 mm$^{-3}$. Time variations of the monitored parameters for one of the experiments are presented in Figure 9.

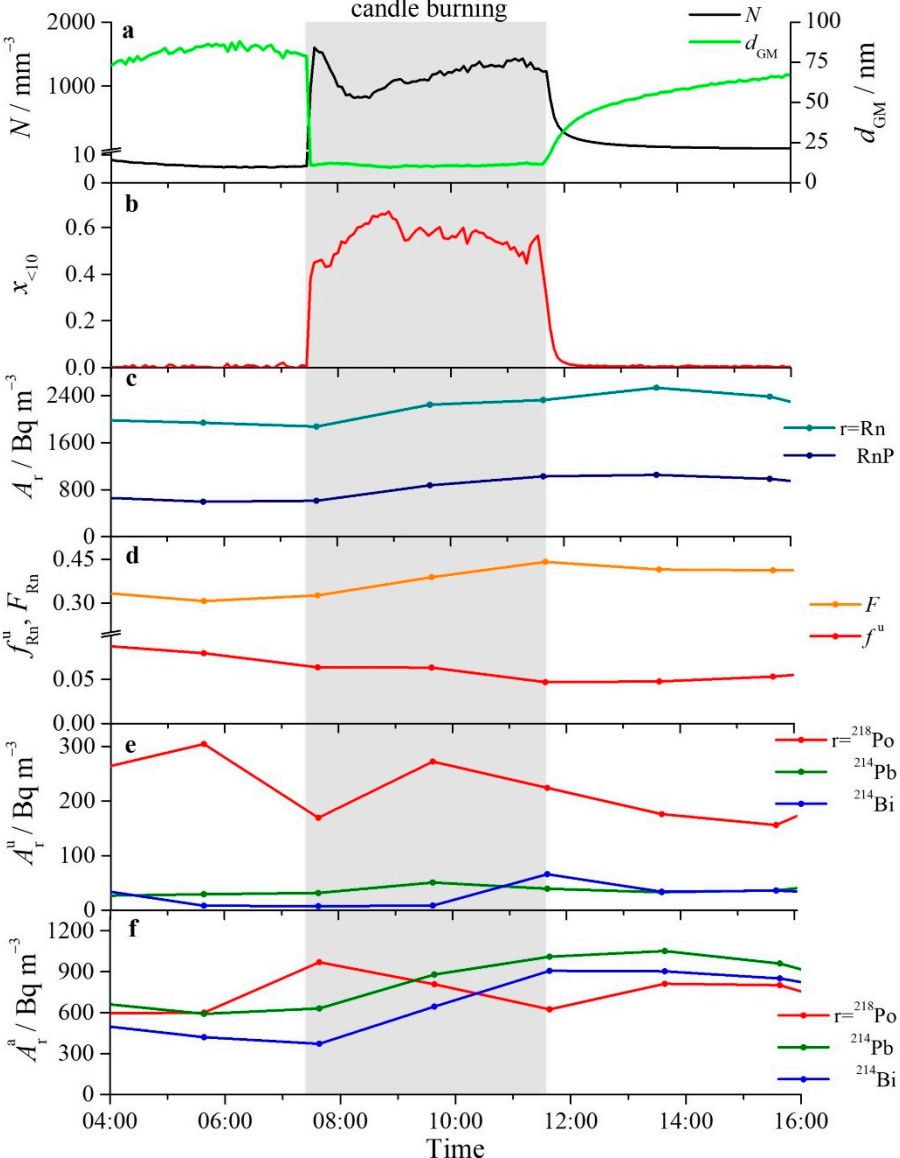

**Figure 9.** Candle burning experiment with a time run of (**a**) the total number concentration of aerosol particles ($N$) and the geometric mean of their diameters ($d_{GM}$), (**b**) the number fraction of particles smaller than 10 nm ($x_{<10}$), (**c**) the activity concentration of radon ($A_{Rn}$) and radon products ($A_{RnP}$), (**d**) the unattached fraction of radon short-lived products ($f_{Rn}^u$) and equilibrium factor between radon and its short-lived products ($F_{Rn}$), (**e**) activity concentrations of the unattached RnP species, $^{218}$Po ($A_{218Po}^u$), $^{214}$Pb ($A_{214Pb}^u$), and $^{214}$Bi ($A_{214Bi}^u$), and (**f**) activity concentrations of the attached RnP species, $^{218}$Po ($A_{218Po}^a$), $^{214}$Pb ($A_{214Pb}^a$), and $^{214}$Bi ($A_{214Bi}^a$).

Soon after lighting the candle, $N$ increased from 5.8 mm$^{-3}$ to 1600 mm$^{-3}$, and $d_{GM}$ decreased from 85 nm to 11 nm (Figure 9a), a similar size as that obtained recently [238]. In contrast to smoking, mostly particles smaller than 10 nm were emitted, as seen in Figure 8b (presenting the number concentrations of particles in the same size windows as in the previous section). After its maximum, $N$ dropped by about 500 mm$^{-3}$ (Figure 9a), primarily due to a decrease in concentrations of both 9.3 nm and 29.6 nm particles (Figure 8b), presumably due to not constantly burning and not due to particle coagulation because the appearance of larger particles (e.g., 68.4 nm) was delayed and, in addition, their concentration did not follow the trend of steady concentration increases of smaller particles (Figure 8b). Two hours after burning, <20 nm particles had almost completely disappeared from the size distribution (Figure 10).

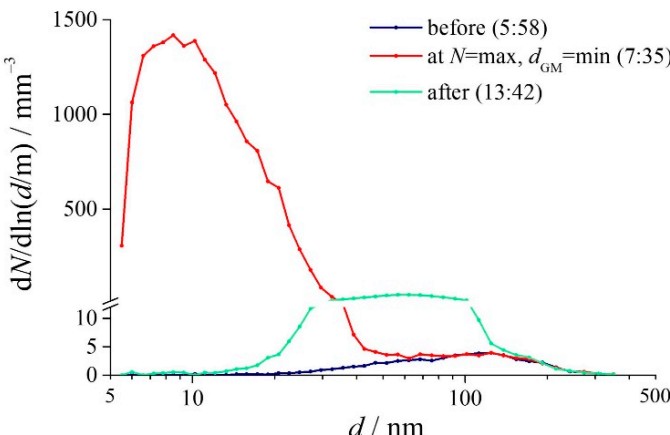

**Figure 10.** Candle burning experiment: number size distribution of aerosol nanoparticles before candle burning (at 5:58), at $N$ = maximum and at $d_{GM}$ = minimum (at 7:35), and after cigarette smoking (at 13:42).

During candle burning, both $A_{Rn}$ and $A_{RnP}$ were slightly enhanced (a shadowed region in Figure 9c), while, as expected [56,101,176], $F_{Rn}$ was increasing and $f_{RnP}^{u}$ was decreasing. $F_{Rn}$ changed from 0.31 to 0.41 (factor of increase of 1.33) and $f_{RnP}^{u}$ from 0.08 to 0.05 (Figure 9d). The decrease in $f_{RnP}^{u}$ originated mostly from the increase in concentrations of the attached $^{214}$Pb and $^{214}$Bi species (Figure 9e,f). The initial $A_{218Po}^{u}$ decrease was compensated by the $A_{218Po}^{a}$ increase and vice versa; later, the $A_{218Po}^{u}$ increase was compensated by the $A_{218Po}^{a}$ decrease. For the above $f_{RnP}^{u}$ values of 0.08 and 0.05, Equation (16) would give values of 7.54 and 7.04, respectively, i.e., a factor of decrease of 1.07 for $f_{DC}$. Hence, $F_{Rn}$ increase would raise the effective dose by a factor of 1.33, and for $f_{RnP}^{u}$, a drop would reduce it by a factor of 1.07, leading to an overall increase of 1.24. Thus, during candle burning in a room, a person would be heavily exposed to nanoparticles and, in addition, would receive about 25% of a higher effective dose than without candle burning.

The candle burning was long enough to show the impact of aerosol characteristics on the behaviour of $f_{RnP}^{u}$ more clearly than in the much shorter smoking experiment. However, based on Figure 9b with a very high fraction of the <10 nm particles (associated with the unattached RnP), an increase in $f_{RnP}^{u}$ would be expected instead of a decrease. An explanation of this false anticipation is presented in Figure 11, showing the ratios of the number concentrations and of the surface area concentrations of the <10 nm particles versus >10 nm particles. Although the number concentration of smaller particles during candle burning was up to twice as much as that of larger ones, their surface area concentration was only from 4 to 14% of that of bigger particles. Hence, the interaction of RnP species with bigger aerosol particles was preferred, thus leading to a $f_{RnP}^{u}$ decrease. The role of the surface area of aerosol particles in governing the $f_{RnP}^{u}$ behaviour [231,235] was also reported recently by Vargas Trassierra et al. [108].

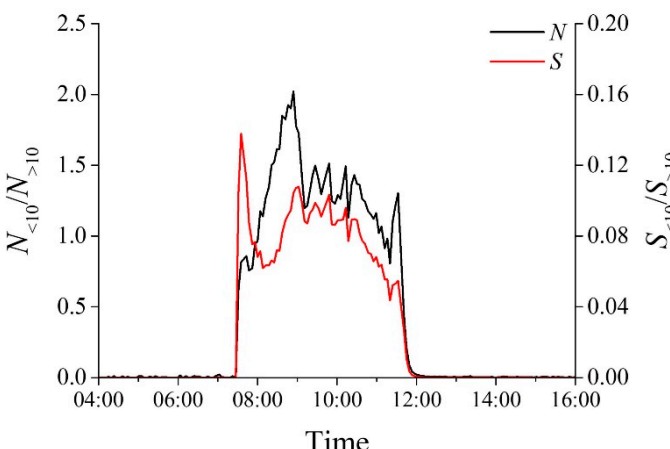

**Figure 11.** Candle burning experiment: the ratios of the number concentrations of the <10 nm and >10 nm particles ($N_{<10}/N_{>10}$) and of the surface area concentrations of the <10 nm and >10 nm particles ($S_{<10}/S_{>10}$).

## 7. Conclusions

Exposure of the population to radon and thoron and their short-lived products has been one of the focal concerns of the national and international institutions responsible for human health for decades. Their concern is based on radon and thoron surveys carried out in a large number of countries worldwide. In many countries, systematic measurements were conducted at the national level or at least in selected areas. Although the priority was given to dwellings, workplaces (other than uranium mines) have not been ignored. Radon levels indoors were complemented by levels in soil gas, outdoor air, and water, with the ultimate aim of assessing the population's radiation dose and keeping it sufficiently low. Findings from the laboratory experiments have enriched the results of field measurements. Epidemiological studies have played a significant role. A considerable effort has also been devoted to modelling the pathways of radon and thoron and their products from their sources, entry into buildings, dynamics in indoor air, deposition in lungs, and the effects on health. The database for indoor radon and its products is far more extensive than that for thoron and its products, although this gap has narrowed recently, and this trend should continue.

Efforts to harmonise the results of different radon groups worldwide should also be continued. The goal of preparing a radon atlas for continents other than Europe, or even a global one, should not be considered too optimistic.

It should also be borne in mind that the exposure of the population to radon and thoron and their products in a region or a country is not fixed and may change due to changes in living–working habits or construction characteristics (e.g., passive buildings with better thermal insulation). If this happens, radon should be re-checked.

In recent decades, the intensity of massive radon measurements in the indoor air of dwellings and public buildings has been enhanced, and rich knowledge of radon dynamics under real living and working conditions has been gained. Meanwhile, the extent of fundamental research on radon under well-defined experimental conditions in radon chambers has not increased proportionally or has even decreased. This trend should be reversed in the future because the results obtained in the laboratory may, firstly, serve as helpful guidance in designing measurement protocols in field radon campaigns and, secondly, should be a prerequisite for interpreting the data measured therein. Laboratory results on the role of aerosol characteristics in the dynamics of radon products would be beneficial, particularly their partitioning between attached and unattached forms. This would shed more light on the radon dosimetry approach and could eventually change or improve it. Sustainable indoor air quality management requires a balance between real-world measurements and laboratory-based studies.

**Funding:** This research was funded by the Slovenian Research and Innovation Agency (research core funding No. P1-0143, Cycling of substances in the environment, mass balances, modelling of environmental processes and risk assessment), the Ministry of Higher Education, Science and Innovation, and the European Union from the European Regional Development Fund (Development of research infrastructure for the international competitiveness of the Slovenian RRI space–RI-SI-EPOS).

**Conflicts of Interest:** The author declare no conflict of interest.

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
