# Peer review of "Radon and Its Short-Lived Products in Indoor Air: Present Status and Perspectives"

_sustainability, doi:10.3390/su16062424_

Round 1

Reviewer 1 Report

Comments and Suggestions for Authors

Exposure to radon and thoron is the second most important cause of lung cancer after cigarette smoking. Therefore, many studies on radon and thoron and their short-lived progeny radioactive aerosols have been carried out in the world, such as the generation of radon and thoron, the formation of radioactive aerosols, the measurement methods and techniques, radioactive concentrations and their change with time and environmental factors, internal exposure dose due to inhalation, etc. Many research papers including review articles have been published. This paper is a comprehensive review. The formulas for calculating radioactive concentrations of radon, thoron and their progeny aerosol indoor air under the equilibrium condition are given. The measured concentrations and their changes of indoor radon, thoron and the radioactive aerosol in indoor air with time, meteorological conditions and human activities are summarized. It is a pity that there is no introduction to the progress on the methods and techniques about the concentration of radon, thoron and their progeny radioactive aerosols.

The article is long and time-consuming to read. It is suggested that in the summary of Section 7, in addition to the qualitative description, the measured values or ranges of the radioactivity concentrations of radon, thoron and radioactive aerosol with typical time periods, meteorological conditions and human activities should be clearly listedwhich is more interesting and useful to readers.

Author Response

To Reviewer 1

The author wishes to express gratitude to the reviewer for a thorough examination of the lengthy text, positive feedback, and constructive comments.

There are two issues expressed in the review:

  1. R: It is a pity that there is no introduction to the progress on the methods and techniques about the concentration of radon, thoron and their progeny radioactive aerosols.

A: I agree that this chapter would be helpful, and I thought about it. However, this is a vast field, and a clear and systematic review would considerably lengthen the already long text. So, I preferred to focus on the chapters I present.

  1. R: The article is long and time-consuming to read. It is suggested that in the summary of Section 7, in addition to the qualitative description, the measured values or ranges of the radioactivity concentrations of radon, thoron and radioactive aerosol with typical time periods, meteorological conditions and human activities should be clearly listed, which is more interesting and useful to readers.

A: I read Section 7 (Conclusion) a few times to include the measured values as you suggested. But, this would completely change the concept of the conclusion, where I briefly summarised the state of research today and in which direction, in my opinion, it would be helpful to continue.

This comment refers primarily to Section 6, where the results of several studies, each related to a specific location, season, and activity, are discussed. In the end, it would be hard to extract values from this; I guess the ranges are too wide to summarize meaningfully. For this, the conclusion should be significantly extended, which would, unfortunately, change my concept.

If I were summarizing the results of several similar experiments, I would certainly include the values in the conclusion, but in this case, I did not; I hope you accept my arguments.

Reviewer 2 Report

Comments and Suggestions for Authors

Report on Sustainability-2828391

This ms present good review of radon/thoron progeny concentration and their behavior in indoor air.

MS is well and clearly written and I recommend it for publication after minor corrections

Some short survey of radon progeny measuring techniques is welcome here.

Line  98, Written ‘”218Po and 216Po become neutralised “ change to “218Po and 216Po are formed as positively charged and become neutralised ..”

Lines 116, 117. Phrase “order of magnitude 100 eV” was repeated two time. Delete second appearance.

Lines 118, 119. “Unattached  and attached RnP and TnP species behave as radioactive aerosols in the air”. This is not completely correct because unattached progeny behaves different in some processes. Actually, they behave very different during breathing processes, although the basic processes are the same.

Line 127. Written “and coarse “.  Please do not use “coarse” in this context.  It should be attached mode. According to some authors, attached mode consists of three modes, nucleation, accumulation and coarse mode. It is confusing here to use coarse. Also instead Vdc  use vda. It has already done in Table 3 caption.

Line 214. Written “Ambient air is an aerosol with suspended particulate matter”.   Cannot say that air is an aerosol. Rewrite this sentence.

Line 262. “addition, RnP appeared in the” change to “addition, attached RnP appeared in the..”

Line 256 to 278. Discussion about size of unattached and attached progeny. One should mention that there are typical values of those parameters adopted in lung dosimetry calculation by Marsh and Birchall.

Lines 422-423. As a source of radon author can add used tap water and gas.

Fig. 5. Log axis on ordinate hide something. Please try to use linear scale. If it is not possible try to present data for radon  and thoron on separate graphs.

Author Response

To Reviewer 2

The author wishes to express gratitude to the reviewer for thoroughly examining the lengthy text, giving positive feedback, and providing constructive comments, all of which contribute to improving the scientific quality of the manuscript.

There is one issue expressed in the review, and several corrections or comments refer to the exact lines of the manuscript.

R: Some short survey of radon progeny measuring techniques is welcome here.

A: I agree that this chapter would be helpful, and I thought about it. However, this is a vast field, and even a short systematic review would considerably lengthen an already long text. So, I preferred to focus on the chapters I presented.

R: Line 98, Written '‘" ”218Po and 216Po become neutralised "“ change to "“218Po and 216Po are formed as positively charged and become neutralised .."”

A: Yes, I added.

R: Lines 116, 117. Phrase "“order of magnitude 100 eV"” was repeated two time. Delete second appearance.

A: Yes, I deleted it.

R: Lines 118, 119. "“Unattached  and attached RnP and TnP species behave as radioactive aerosols in the air"”. This is not completely correct because unattached progeny behaves different in some processes. Actually, they behave very different during breathing processes, although the basic processes are the same.

A: I changed the above sentence to "“Attached RnP and TnP species behave as other radioactive and non-radioactive aerosol particles."”

R: Line 127. Written "“and coarse "“.  Please do not use "“coarse"” in this context.  It should be attached mode. According to some authors, attached mode consists of three modes, nucleation, accumulation and coarse mode. It is confusing here to use coarse. Also instead Vdc  use vda. It has already done in Table 3 caption.

A: Yes, I changed.

R: Line 214. Written "“Ambient air is an aerosol with suspended particulate matter"”.   Cannot say that air is an aerosol. Rewrite this sentence.

A: I changed the above sentence to "“Ambient air is a suspension of fine solid particles or liquid droplets."”

R: Line 262. "“addition, RnP appeared in the"” change to "“addition, attached RnP appeared in the.."”

A: Yes, I added.

R: Line 256 to 278. Discussion about size of unattached and attached progeny. One should mention that there are typical values of those parameters adopted in lung dosimetry calculation by Marsh and Birchall.

A: I added the following sentence to the end of the paragraph: "“The typical values of those parameters have been adopted in lung dosimetry calculation by Marsh and Birchall [106].

The reference (I hope it is the correct one):

Marsh, J.W.; Birchall, A. Sensitivity analysis of the weighted equivalent lung dose per unit exposure from radon progeny. Radiat. Prot. Dosim. 2000, 87, 167–178.

R: Lines 422-423. As a source of radon author can add used tap water and gas.

A: Yes, I added.

R: Fig. 5. Log axis on ordinate hide something. Please try to use linear scale. If it is not possible try to present data for radon and thoron on separate graphs.

A: Yes, I used the linear scale and inserted new graphs in Fig 5. In the discussion (lines 672‒676), I added the factors of increase.

In the revised manuscript, all changes are highlighted.

Reviewer 3 Report

Comments and Suggestions for Authors

The paper's writing style is clear, concise, and engaging, making it suitable for a wide range of audiences. You have successfully synthesized complex ideas and presented them in an informative and compelling way. The paper is well-organized, and the citations are comprehensive and relevant.

I am writing to strongly recommend that the review "Radon and its short-lived products in indoor air: present status and perspectives" be accepted for publication. After thoroughly reviewing the manuscript, I am confident that this well-written and meticulously researched paper will contribute significantly to the field of indoor environments. Based on my assessment, I believe that publishing this paper will make a significant contribution to academic discourse in indoor environments and will be of great interest to Sustainability readers. As a result, I strongly recommend that the paper be accepted for publication in your esteemed journal.

Author Response

To Reviewer 3

I would like to express my gratitude to the reviewer for their meticulous examination of the extensive text. I am delighted to receive a positive opinion and a recommendation for publication. Crafting a peer-reviewed article demands a systematic review of the literature and significant time investment, making this recognition an excellent incentive for future endeavours.